# Quantitative somatosensory assessments in patients with persistent pain following groin hernia repair: A systematic review with a meta-analytical approach

**Akhmedkhan Dubayev**[1]*, **Elisabeth Kjær Jensen**[1], **Kenneth Geving Andersen**[2], **Martin F. Bjurström**[3], **Mads U. Werner**[1]

1 Multidisciplinary Pain Center, Neuroscience Center, Copenhagen University Hospitals - Rigshospitalet, København, Denmark, 2 Department of Anesthesia and Intensive Care, Copenhagen University Hospitals - Hvidovre Hospital, Hvidovre, Denmark, 3 Department of Surgical Sciences, Uppsala University, Uppsala, Sweden

* akhmedkhan.dubayev@hotmail.com

## Abstract

### Objectives

Quantitative sensory testing (QST) provides an assessment of cutaneous and deep tissue sensitivity and pain perception under normal and pathological settings. Approximately 2–4% of individuals undergoing groin hernia repair (GHR) develop severe persistent postsurgical pain (PPSP). The aims of this systematic review of PPSP-patients were (1) to retrieve and methodologically characterize the available QST literature and (2) to explore the role of QST in understanding mechanisms underlying PPSP following GHR.

### Methods

A systematic literature search was conducted from JAN-1992 to SEP-2022 in PubMed, EMBASE, and Google Scholar. For inclusion, studies had to report at least one QST-modality in patients with PPSP. Risk of bias assessment of the studies was conducted utilizing the Newcastle Ottawa Scale and Cochrane's Risk of Bias assessment tool 2.0. The review provided both a qualitative and quantitative analysis of the results. A random effects model was used for meta-analysis.

### Results

Twenty-five studies were included (5 randomized controlled trials, 20 non-randomized controlled trials). Overall, risk of bias was low. Compared with the contralateral side or controls, there were significant alterations in somatosensory function of the surgical site in PPSP-patients. Following thresholds were significantly increased: mechanical detection thresholds for punctate stimuli (mean difference (95% CI) 3.3 (1.6, 6.9) mN (P = 0.002)), warmth detection thresholds (3.2 (1.6, 4.7) ˚C (P = 0.0001)), cool detection thresholds (-3.2 (-4.9, -1.6) ˚C (P = 0.0001)), and heat pain thresholds (1.9 (1.1, 2.7)

**Data Availability Statement:** All relevant data are within the paper and its Supporting information files.

**Funding:** The author(s) received no specific funding for this work.

**Competing interests:** The authors have declared that no competing interests exist.

**Abbreviations: BMI**, Body Mass Index; **CDT**, Cool Detection Threshold; **CPT**, Cold Pain Threshold; **DFNS**, German Research Network on Neuropathic Pain; **DN4**, Douleur Neuropathique 4; **GHR**, Groin Hernia Repair; **HADS**, Hospital Anxiety and Depression Scale; **HPT**, Heat Pain Threshold; **IENFD**, Intraepidermal Nerve Fiber Density; **MDT**, Mechanical Detection Threshold; **MPT**, Mechanical Pain Threshold (punctate); **NOS**, Newcastle-Ottawa Scale; **NRS**, Numeric Rating Scale; **P**, = P-value; **PCS**, Pain Catastrophizing Scale; **PPSP**, Persistent Postsurgical Pain; **PPT**, Pressure Pain Threshold (blunt); **QST**, Quantitative Sensory Testing; **RCT**, Randomized Controlled Trial; **RoB 2.0**, Cochrane's Risk of Bias Assessment tool 2.0; **SF-36**, Short Form Health Survey; **SF-MPQ**, Short-form McGill Pain Questionnaire; **STHS**, Suprathreshold Heat Stimulus; **VAS**, Visual Analog Scale; **WDT**, Warmth Detection Threshold.

˚C (P = 0.00001)). However, the pressure pain thresholds were significantly decreased (-76 (-123, -30) kPa (P = 0.001)).

## Conclusion

Our review demonstrates a plethora of methods used regarding outcome assessments, data processing, and data interpretation. From a pathophysiological perspective, the most consistent findings were postsurgical cutaneous deafferentation and development of a pain generator in deeper connective tissues.

## Trial registration

CRD42022331750.

## 1. Introduction

### 1.1 Background

**1.1.1 Persistent postsurgical pain following groin hernia repair.** Groin hernia repair (GHR) is a common surgery performed in more than 20 million patients worldwide every year [1]. Persistent postsurgical pain (PPSP) following GHR is a well-known medical complication [2] and efficacious management of patients with PPSP remains a major challenge for the healthcare profession [3]. The IASP (the International Association for the Study of Pain [ICD-11]) criteria for chronic post-surgical or posttraumatic pain are "a chronic pain that develops or increases in intensity after a surgical procedure or a tissue injury and persists beyond the healing process, i.e., at least three months after the surgery or tissue trauma" [4]. More elabo-rate criteria have previously been proposed [5]. The condition can significantly impair the physical and psychosocial functions of the individual, and a conservative estimate is that 2% of patients undergoing groin hernia repair will be affected by PPSP [6, 7]. The prevalence of PPSP is primarily contingent on the respective surgical procedure, whilst patient-related pre-surgical factors also affect the frequency and severity of PPSP [1, 8, 9].

The anatomical region in which the surgery is performed [10] is complex, with a high degree of vascularization, dense nerve fiber innervation, and several peripheral nerves travers-ing the region [10]. Additionally, the region has a significant role in posture control, locomo-tion, as well as reproductive functions [10–12]. The three primary causes of chronic pain after groin hernia repair are inflammatory processes caused by foreign materials, formation of a meshoma or the development of neuropathic pain, e.g., nerve injury, through nerve transec-tion, compression or entrapment or devascularization [7].

One of the key investigative tools applicable in PPSP is QST (quantitative somatosensory testing), which has been used extensively in the research of pathophysiological mechanisms [7, 13]. A thorough examination of the somatosensory characteristics of patients, with the use of QST, could further decipher pathways underlying chronic pain, which might guide the thera-peutic management of the condition [14]. However, no systematic nor methodological review of the findings pertaining to the use of QST in patients suffering from PPSP following GHR has been published yet.

In relation to the treatment of PPSP following GHR, re-surgery, either in terms of meshect-omy, or selective or triple neurectomy, is quite effective in treating a subset of patients [7, 15, 16], though the need for improvement of non-interventional treatment of PPSP is still an important factor to consider [17].

**1.1.2 Quantitative sensory testing (QST).** QST is a standardized, non-invasive psycho-physical testing procedure where the individual is exposed to various graded stimulation modalities (e.g., thermal and mechanical). The stimulus evoked responses are quantified by the individual in terms of sensory detection and pain thresholds. [18, 19]. An example of a thorough QST protocol is the standardized protocol provided by the DFNS (German Research Network on Neuropathic Pain).

Through performance of the QST procedure, it is possible to evaluate and quantify the somatosensory profile of individuals by assessing the function of different nerve fiber types in the somatosensory system [14].

**1.1.3 Aims of this study.** The aims of this systematic review were: *First*, to retrieve and methodologically characterize the available literature related to QST in patients with PPSP following GHR. *Second*, to explore the role of QST in understanding mechanisms underlying PPSP following GHR.

From a pathophysiological perspective, the review may facilitate evaluation of the diagnostic efficiency of QST methods and, consequentially, improved treatment paradigms.

## 2. Materials and methods

The review was conducted in accordance with PRISMA guidelines [20] and was registered in the PROSPERO international register of systematic reviews (CRD42022331750). The PRISMA 2020 checklist is available as S1 Checklist.

### 2.1 Eligibility criteria

This review included studies concerning the use of QST on human subjects who had undergone GHR and were subsequently affected by PPSP. There were no exclusion criteria related to surgical technique; both open and laparoscopic surgeries were considered. QST was defined as an examination procedure of the groin using a quantifiable approach. Studies were eligible if a technique of somatosensory examination of the groin using QST had been used (cf. 1.1.2.). Studies were included if at least one standard mechanical (MDT, MPT, PPT) or one thermal (WDT, CDT, HPT, CPT) modality related to QST was described in the study. No restriction regarding age was implemented in the literature search. Studies were included if the post-surgical assessment period was at least three months. Eligible studies were published in English or German language, during the period JAN-1992 to SEP-2022.

### 2.2 Systematic literature search

A systematic literature search was conducted using several online databases (PubMed, EMBASE, and Google Scholar). Furthermore, the authors searched PROSPERO and PubMed for published or ongoing reviews with the following search string:

'Chronic Pain'[MeSH Terms] OR 'pain, postoperative'[MeSH Terms] OR chronic pain*[Text Word] OR postoperative pain*[Text Word] OR persistent pain*[Text Word] OR postsurgical pain*[Text Word]') **AND**

('Hernia, Inguinal'[Mesh] OR 'Herniorrhaphy'[Mesh] OR groin hernia repair*[Text Word] OR inguinal hernia repair*[Text Word] OR inguinal herniorrhaphy*[Text Word] OR groin hernia herniorrhaphy*[Text Word] OR inguinal hernia*[Text Word]') **AND**

('Sensory Thresholds'[Mesh] OR 'Pain Threshold'[Mesh] OR 'Pain Measurement'[Mesh] OR quantitative sensory test*[Text Word] OR QST [Text Word] OR somatosensory test* [Text Word] OR sensory profile*[Text Word] OR sensory threshold*[Text Word] OR pain

threshold*[Text Word] OR warmth detection test*[Text Word] OR cold detection test* [Text Word] OR heat pain threshold*[Text Word] OR cold pain threshold*[Text Word] OR pressure pain threshold*[Text Word] OR quantitative sensory test*[Text Word]')

The final systematic search strategy and search paradigm, including citation tracking, were determined with the help of a research librarian. The search strategy was not limited to a specific study type. Randomized controlled trials (RCTs), cohort studies, and systematic reviews were included to secure as much of the relevant literature and data on the topic as possible. To further maximize the effectiveness of the search, manual searching was also performed in the reference lists of full-text studies to capture publications that might be overlooked in the online search.

## 2.3 Data extraction

**2.3.1 Study selection process.**   After conducting the search using the MeSH terms and Text Words, the initial screening was based on the titles. The subsequent selection of studies was determined by reviewing abstracts of these studies, followed by a final full-text screening. The respective studies obtained through the search were independently screened by three authors (AD, EKJ, MW) to identify the literature which met the eligibility criteria. In case of ambiguities, the senior author (MW) made the final decision.

**2.3.2 Data extraction.**   A data extraction sheet was produced with the purpose of systema-tizing relevant information extracted from each of the included studies. The extracted infor-mation included study design, number of eligible participants, demographics, QST protocol details, QST variables generated in the studies (e.g., thermal and mechanical detection and pain thresholds), and other relevant outcome measures. To establish a consistent and thorough data extraction, the three authors extracted and analyzed the data. No attempts were made to contact the respective study authors.

## 2.4 Assessing the risk of bias

The Newcastle Ottawa Scale (NOS) [21] and the Cochrane Risk of Bias Tool 2.0 (RoB 2.0) [22], were used to evaluate the methodological quality and risk of bias of non-RCT's and RCT's, respectively. The quality assessment was conducted independently and discussed by the three authors, and the senior author made the final decision in case of ambiguities.

The NOS is a tool with the purpose of assessing the risk of bias and overall qualities of non-RCTs. The tool utilizes a "star system" to delegate points based on three main perspectives: the selection of study groups, comparability of study groups, and the ascertainment of either the exposure or outcome of interest (depending on whether the study in question is a case-control study or cohort study, respectively). Each main domain contains further questions, totaling eight different assessment areas. A maximum of nine stars/points can be allotted to the study in question [21].

The RoB 2.0 is an instrument to assess risk of bias and quality of RCTs [22]. The tool focuses on a set of domains related to multiple characteristics of trial design, conduct, and reporting. Each main domain is associated with a series of signaling questions aiming to report on specific features of the trials. The tool contains an algorithm providing a conclusive assess-ment of the risk of bias. Studies are categorized as being either "Low" or "High" in terms of risk of bias or that there are "Some concerns".

## 2.5 Strategy for data synthesis and data analysis

Data analysis was undertaken using Review Manager 5.4.1 [23] for the creation of forest plots. In total, data from 7 different QST modalities were used for analysis. For pooled analyses, data

from 6 studies of punctate mechanical detection threshold (MDT) and mechanical pain threshold (MPT) [12, 24–28], data from 11 studies of warmth detection threshold (WDT) and heat pain threshold (HPT), data from 9 studies of cool detection threshold (CDT) [12, 24–31] and 6 studies of cold pain threshold (CPT) [12, 24, 26–28, 31], and data from 7 studies of blunt pressure pain threshold (PPT) [12, 24, 25, 27–30], were used. Outcome data were found qualified for analysis if the data were presented as continuous data (mean or median) and if the study included a comparison of data (surgical side *vs.* non-surgical side). Between the studies, the data for a specific modality would differ in presentation. For example, some studies presented MDT/MPT-values on a logarithmic scale, while other studies solely presented raw data, means, or medians. As such, to secure homogeneity between outcome data, all values were transformed to represent the same distribution. Data for MDT/MPT-values were log-transformed, while remaining forest plots contain raw data (WDT, CDT, HPT, CPT, PPT). For the data in intervention studies, only baseline values, i.e., pre-intervention values, were used for the analysis. The purpose of this was to avoid comparing outcome data affected by the respective interventions with data from non-interventional studies. In the forest plots, results on the left part of the abscissa, which represented a decrease in thresholds from the affected side to control side, were labeled "gain of sensory function". Correspondingly, the right part of the abscissa represented an increase in thresholds indicating a "loss of sensory function" [32]. The analyses were conducted using a random effects model. Furthermore, standard tests of heterogeneity were performed. Data are, unless otherwise stated, indicated as mean (95% CI). For the interpretation of results in this review, a P-value of 0.01 was set to minimize the likelihood of a type 1 error due to multiple comparisons.

## 3. Results

### 3.1 Literature search

A PRISMA flow diagram is presented in Fig 1. The final search resulted in 1,105 records. After the subsequent exclusion of 11 duplicates, a total of 1,094 records were screened. After the

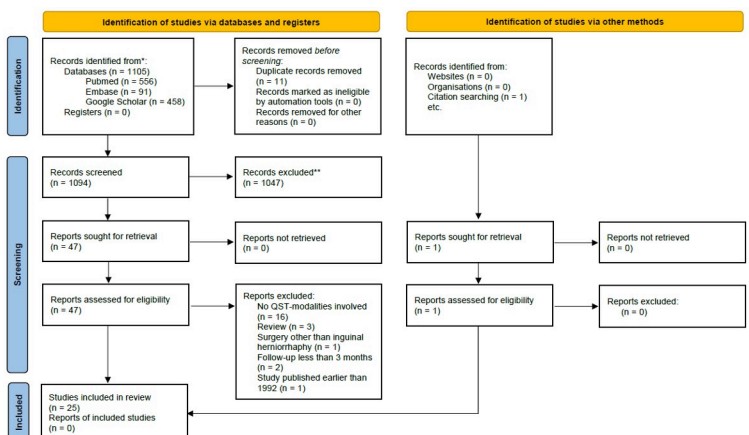

**Fig 1. PRISMA 2020 flow diagram for new systematic reviews which included searches of databases, registers and other sources.** *Consider, if feasible to do so, reporting the number of records identified from each database or register searched (rather than the total number across all databases/registers). **If automation tools were used, indicate how many records were excluded by a human and how many were excluded by automation tools. *From*: Page MJ, McKenzie JE. Bossuyt PM, Boutron l, Hoffmann TC, Mulrow CD, et al. The PRISMA 2020 statement: an updated guideline for reporting systematic reviews. BMJ 2021; 372:n71. doi: 10.1136/bmj.n71. For more information, visit: http://www.prisma-statement.org/.

initial screening of titles and abstracts, 1,046 records were excluded resulting in 47 reports assessed for eligibility. Further, 23 studies were excluded by full-text screening. A single study was identified through citation tracking, resulting in a total yield of 25 studies included in the review.

## 3.2 Study characteristics

The characteristics of the studies are presented in Table 1. Twenty of the studies that met the inclusion criteria were non-RCTs [7, 10, 13, 15, 18, 24–28, 31, 33–41]; five of these were cross-sectional studies [18, 26, 27, 35, 37]. The other non-RCTs were cohort studies. The remaining 5 studies were RCTs [8, 9, 29, 30, 42]. The total number of study subjects was 1,984, of whom 153 were from RCTs. Female study subjects comprised 114 of the participants (6 from RCTs, 108 from non-RCTs), corresponding to 6% of the total participant population. All but two studies [25, 27] reported additional non-QST related outcomes, most importantly pain assessment through interview or various questionnaires (e.g., IPQ (Inguinal Pain Questionnaire), DN4 (douleur neuropathique 4), SF-MPQ (short-form McGill Pain Questionnaire), and generic psychosexual evaluation questionnaires). Additionally, 10/25 studies [7, 10, 12, 15, 24, 29, 30, 33, 36, 38] included psychological evaluation of the participants by utilizing the HADS (Hospital Anxiety and Depression Scale) and/or the PCS (Pain Catastrophizing Scale). Quality of life aspects were assessed in 5/25 studies [13, 15, 28, 34, 36] using the SF-36 (the 36-item Short Form Health Survey).

Interventions in the RCTs were purified capsaicin instillation [42], nerve conduction block with 1% and 2% lidocaine [8], use of 5% lidocaine patch [29], use of 8% capsaicin patch [30], and ultrasound-guided tender point blockade with 0.25% bupivacaine [9]. A total of 11/25 studies [12, 18, 24–31, 33] were found qualified for statistical analysis. The authors used a pragmatic process to select the papers that presented extractable data for the analysis. For example, some studies only presented data as differences (delta-values), whilst the statistical analysis required absolute values. Another reason for a study not presenting data qualified for analysis could simply be that the QST-variables belonged to secondary or even tertiary outcomes. Of the 11 studies used for analysis, 2 were RCTs, and 9 non-RCTs. Raw data used for analysis from the 2 RCTs were not found in the studies, as data were only presented as differences across an intervention; however, one of the review authors (MW) was also one of the authors in the RCTs, and raw data were obtained, and used for analysis. The results from the statistical analyses are illustrated in the forest plots (Fig 2A–2G).

**3.2.1 Demographics.** The demographics of included study participants are presented in Table 2. The mean age (SD) of all participants was 49.9 (10.3) yrs. One study [12] solely reported the range of age and was therefore not taken into consideration when calculating the mean age of study subjects. A single study stood out in terms of age since the study was conducted in children with the mean (range) age of 8 years (6 months—12 years) [37]. Calculating age only from adult studies, the mean (SD) was 51.2 (10.1) yrs. The BMI of included participants was reported in 7/25 studies [9, 13, 28–30, 33, 35] with a mean (SD) of 26.5 (1.8) kg/m$^2$.

## 3.3 QST characteristics

**3.3.1 Randomized controlled trials.** Thermal sensitivity evaluated by sensory mapping with a metal thermo-roller was assessed in all 5 RCTs. Mechanical hypersensitivity (allodynia) for brush and hypo- or hyperalgesia for punctate stimulation was additionally assessed with sensory mapping in one of these studies [42].

MDT and MPT were included as QST-modalities in one RCT [42] using 17 progressively rigid polyamide monofilaments ranging in bending force from 0.1 to 2941 mN.

**Table 1. An overview of study characteristics.**

| First author, publication year, ref# | Study design | Number M/F | Objectives | Outcome measures (non QST-related) | Active intervention | Placebo intervention | Persistent pain criteria | Results (non-QST outcomes) | Follow-up after surgery |
|---|---|---|---|---|---|---|---|---|---|
| **Randomized Controlled Trials** | | | | | | | | | |
| Aasvang EK, 2010 [42] | RCT | 41/0 | Evaluation of sensory function after capsaicin wound installation | S-QBPA | Purified capsaicin instillation | Pure water installation | NR | S-QBPA→ | Follow-up at 2.5 yrs |
| Bischoff JM, 2012 [8] | RCT | 22/2 | To investigate the effects of ilioinguinal and iliohypogastric nerve block with lidocaine | P-R, P-A, P-P, SPID | Nerve conduction block (lidocaine) | Saline (0.9%) | NRS >6 (0–10) >6 mos | NR | >6 mos since surgery to be included |
| Bischoff JM, 2013 [29] | RCT | 21/0 | To investigate the effect of lidocaine patch | P-R, P-A, P-P, SPID, SI (DSIS). NP (S-LANSS), psychometrics (PCS, HADS), IENFD | Lidocaine 5% patch | Dermal patch | NRS >6 (0–10) >6 mos | SI→, NP→, HADS→, PCS→, IENFD↓ (pain side vs non-pain side) | >6 mos since surgery to be included. MDP = 47 mos for all subjects |
| Bischoff JM, 2014 [30] | RCT | 42/4 | Investigation of the efficacy of a capsaicin 8% cutaneous patch | P-R, P-A, P-P, SPID, SI (DSIS). NP (S-LANSS), psychometrics (PCS, HADS), IENFD | Capsaicin 8% cutaneous patch | Dermal patch | NRS ≥5 (0–10) >6 mos | SI→, NP→, HADS→, PCS→, IENFD$_{PS}$↓, IENFD$_{I-GRP}$→ | >6 mos since surgery to be included. MDP$_I$ = 37 mos, MDP$_P$ = 39 mos |
| Wijaysinghe N, 2016 [9] | RCT | 20/0 | To investigate the analgesic effects of local anaesthetic tender point blockade | P-R, P-A, P-P, SPID, sleep diary, pain diary | Ultrasound guided tender point blockade (bupivacaine) | Placebo injection | NRS ≥5 (0–10) >6 mos | SPID→, sleep diary→, pain diary→ | >12 mos since surgery to be included. MDP = 35 mos |
| **Non-Randomized Controlled Trials** | | | | | | | | | |
| Mikkelsen T, 2004 [18] | Cross-sectional study | 72/0 | To evaluate the association between chronic pain and sensory dysfunction in hernia repair patients | S-QBPA, S-QBAA (AAS) | NR | NR | 20/72 had experienced pain, mean VAS = 22 (IQR = 12–30) | S-QBPA: Increased pain in 20/72 pat. S-VAS-PI = 22 (IQR = 12–30) in pain group. S-QBAA: Work/leisure activity affected in 11/72 pt. | For inclusion surgery 6–12 mos prior to testing |
| Aasvang EK, 2007 [12] | Cohort study | 30/0 | To investigate the characteristics of sexual dysfunction after groin hernia surgery | PE-I, HADS | NR | NR | Pain categorized into no pain (VAS = 0), light (VAS = 1–3), moderate (VAS = 4–7) or severe pain (VAS = 8–10). Pain occurring "often" or "always" and rates as VAS >3 is termed "substantial pain". | PE-I: Pain of somatic origin in 8/30 pat. Somatic origin of sexual problems in 2/30 pat. | POP-P: 2.1–2.7 yrs, POP-C: 1.7–3.2 yrs |

(*Continued*)

**Table 1.** (Continued)

| First author, publication year, ref# | Study design | Number M/F | Objectives | Outcome measures (non QST-related) | Active intervention | Placebo intervention | Persistent pain criteria | Results (non-QST outcomes) | Follow-up after surgery |
|---|---|---|---|---|---|---|---|---|---|
| Aasvang EK, 2008 [24] | Cohort study | 46/0 | To investigate neurophysiological changes and pathophysiological mechanisms underlying postherniotomy pain | S-QBPA, HADS | NR | NR | UCGP + PRIEA >3 mos (moderate/ severe pain to be included) | S-QBPA: Pain present constantly or daily and daily activities affected in 73% of pat. HADS: Depression/ anxiety found in 5%/8% of pat. | Surgery >1 yr prior to examination to be included |
| Beldi G, 2008 [34] | Cohort study | 89/7 | To investigate chronic pain and hypoesthesia after inguinal hernia repair | S-QBPA, SF-36 (QoL) | NR | NR | NR | POP: 26.7% of pat in group A, 26.1% of pat in group B, 26.0% of pat in group C. MPV: 1.8 (SD = 1.8) in group A, 2.7 (SD = 2.7) in group B, and 1.9 (SD = 1.6) in group C. | Md follow up: 4.7 yrs |
| Aasvang EK, 2009 [26] | Cross-sectional study | 42/0 | To investigate the correlation between preoperative pain intensity and sensory function in the groin hernia area | S-QBPA | NR | NR | No pain (VAS = 0), light (VAS = 1–3), moderate (VAS = 4–7) or severe pain (VAS = 8–10) | S-QBPA: Daily to weekly pain in 32/42 pat | NR |
| Aasvang EK, 2009 [15] | Cohort study | 20/1 | To investigate the effect of neurectomy and mesh removal on persistent postherniotomy pain | S-QBPA, S-QBAA (AAS), HADS, SF-36 | Mesh removal, triple neurectomy | NR | "Patients with persistent pain after groin hernia surgery and pain-related impairment of everyday functions were included" | S-QBPA: Average preoperative pain at rest = 5 (2–8), 3 mos = 3 (0–5), 6 mos = 2 (0–7). S-QBAA↑, HADS: Depression definite in 1/21 pat, doubtful in 1/21 pat, unlikely in 19/21 pat at 6 mo, SF-36↑ | Surgery >1 yrs prior to examination to be included. Md pain duration 4.2 yrs |
| Kalliomäki ML, 2009 [36] | Cohort study | 168/14 | To present a qualitative analysis of persistent post-herniorrhaphy pain | S-QBPA (IPQ), HADS, SF-36, SUSP | NR | NR | IPQ grade >3 to be recruited | S-QBPA (IPQ): Muscular pain in 11 pat, neuropathic pain in 47 pat., HADS→, SF-36$_{P-GRP}$↑ | Persistent pain ≥6 mos to be included |
| Aasvang EK, 2010 [27] | Cross-sectional study | 40 (sex NR) | To establish normative data on sensory function in pain-free patients > 1 year after groin herniotomy | NR | NR | NR | NR | NR | Surgery >1 yr prior to examination to be included. Md follow-up 2.1 yrs (1.0–2.4) |

(*Continued*)

**Table 1.** (Continued)

| First author, publication year, ref# | Study design | Number M/F | Objectives | Outcome measures (non QST-related) | Active intervention | Placebo intervention | Persistent pain criteria | Results (non-QST outcomes) | Follow-up after surgery |
|---|---|---|---|---|---|---|---|---|---|
| Aasvang EK, 2010 [33] | Cohort study | 442/0 | To identify predisposing factors for persistent postherniotomy pain | S-QBPA, S-QBAA (AAS), HADS, PCS | NR | NR | NRS >3 considered as substantial pain | S-QBPA, S-QBAA: Pain experienced in 117 (26.5%) of pat at 6 mos follow-up, HADS→, PCS→ | Follow-up at 6 mos |
| Aasvang EK, 2010 [25] | Cohort study | 70/0 | To evaluate sensory function in persistent postherniotomy pain patients and pain-free control patients | % | NR | NR | Moderate/severe pain, >3 mos Moderate/severe defined as >3 on NRS (0–10) | NR | Surgery >1 yr prior to examination to be included. Md pain duration 3.2 yrs (1.1–8.3). Md follow-up of control group 2.1 yrs (RNG 1.0–2.4) |
| Linderoth G, 2011 [38] | Cohort study | 11/0 | To describe and classify patients with severe persistent pain after laparoscopic herniorrhaphy | S-QBPA, S-QBAA (AAS), HADS | NR | NR | Severe pain = NRS ≥7, moderate pain = NRS 4–6. Md duration after surgery 2 yrs (range 1–14 yrs) | S-QBPA: Md P-R = 5 (Rng 4–8). S-QBAA: Md AAS-score 52% (Rng 21–67). HADS: 3 pat with HADS-scores above normal range. | Md postoperative duration 2 yrs (1–14) |
| Kristensen AD, 2012 [37] | Cross-sectional study | 73/25 (only three children examined with QST) | To examine the prevalence of chronic pain 6–48 mos after inguinal hernia repair in children | S-QBPA (McGill) | NR | NR | NRS >3 (0–10) | S-QBPA: Postoperative pain in 5/98 children | Mn follow-up 3.2 yrs (SD 1.3) |
| van den Broeke EN, 2013 [40] | Cohort study | 15/0 (8 with and 7 without persistent pain) | To investigate whether enhanced ERP N1 amplitude could be a potential marker for altered cortical sensory processing in patients with persistent postsurgical pain | S-QBPA (DN4) | High frequency electrical Stimulation (HFS) | NR | NR | NR | Surgery performed 6–7 yrs prior to examination |
| Moore AM, 2016 [39] | Cohort study | 51/11 (10 patients selected for QST) | Evaluation of long-term outcomes associated with laparoscopic retroperitoneal triple neurectomy | S-QBPA | Laparoscopic retroperitoneal triple neurectomy | NR | Inguinodynia for >6 mos and significant pain, defined as NRS ≥ 6. | S-QBPA→ | Postoperative inguinodynia ≥6 mos to be included. Data collected on days 0, 1, 90, and 6-mos interval up to 3 yrs after intervention |

(*Continued*)

**Table 1.** (Continued)

| First author, publication year, ref# | Study design | Number M/F | Objectives | Outcome measures (non QST-related) | Active intervention | Placebo intervention | Persistent pain criteria | Results (non-QST outcomes) | Follow-up after surgery |
|---|---|---|---|---|---|---|---|---|---|
| Bjurström MF, 2017 [28] | Cohort study | 9/1 | To evaluate neurophysiological and clinical effects of laparoscopic retroperitoneal triple neurectomy | S-QBPA (McGill, NPQ), SF-36, S-QBAA (AAS), PSQI | Laparoscopic retroperitoneal triple neurectomy | NR | Pain duration ≥6 mos | S-QBPA↑, S-QBAA↑, SF-36↑, PSQI↑ | Inguinodynia ≥6 mos to be included. Data collected at baseline (prior to surgery), 2 weeks, 3 mos and 6 mos after intervention. |
| Bjurström MF, 2017 [13] | Cohort study | 12/1 | To examine the association between sensory mapping, pre- and postoperative QST results in patients undergoing triple neurectomy | S-QBPA (McGill, NPQ), SF-36, S-QBAA (AAS), PSQI | Laparoscopic retroperitoneal triple neurectomy | NR | Pain duration ≥6 mos | S-QBPA↑, S-QBAA↑, SF-36↑, PSQI↑ | Inguinodynia ≥6 mos to be included. Data collected at baseline (prior to surgery), 2 weeks, 3 mos and 6 mos after intervention. |
| Ergönenc T, 2017 [35] | Cross-sectional study | 230/34 | To assess the prevalence of chronic pain after inguinal hernia repair and the effects on quality of life | S-QBPA (IPQ, DN4) | NR | NR | NR | S-QBPA↑ | >3 mos since surgery to be included. |
| Jensen EK, 2019 [7] | Cohort study | 190/14 | To present data in PPHP; To evaluate functional + pain-related outcomes of re-surgery or pharmacotherapy by 5-yr questionnaires | S-QBPA, S-QBAA (AAS), HADS, PCS | NR | NR | NR | S-QBPA↑, S-QBAA↑ | Md follow up: >2 yrs |
| Wheeler DW, 2019 [31] | Cohort study | 18/0 | To assess the feasibility of mechanical hyperalgesia or allodynia as outcomes for assessment of early analgesic efficacy | S-QBPA (NPSI), Spontaneous and Dynamic Pain and Lung Function | NR | NR | NR | S-QBPA: Md. NPSI-score 1 yr after surgery = 1.5 (max score 100). Spontaneous and Dynamic Pain and Lung Function↑ | Subjects were assessed 2 – 4 weeks before surgery, and at 2, 4, 6, 8, 16, and 24 weeks after surgery. |

(*Continued*)

**Table 1.** (Continued)

| First author, publication year, ref# | Study design | Number M/F | Objectives | Outcome measures (non QST-related) | Active intervention | Placebo intervention | Persistent pain criteria | Results (non-QST outcomes) | Follow-up after surgery |
|---|---|---|---|---|---|---|---|---|---|
| Jensen EK, 2021 [10] | Cohort study | 95/0 | To evaluate pain-trajectories in a cohort referred from groin hernia repair-surgeons to a tertiary pain-center. | S-NRS, S-QBAA (AAS), HADS, PCS | NR | NR | NRS >7 | S-NRS, Md (95% CI): 25.8 (24.5–27.0), S-QBAA, Md (95% CI): 12.0 (11.0–13.0), PCS, Mn (95% CI): 24.7 (16.8–32.6), HADS-A, Mn (95% CI): 7.4 (2.8–12.0), HADS-D, Mn (95% CI): 5.6 (0.6–10.7) | |

**AAS** = Activity Assessment Scale; **B-WU** = Brush windup; **c** (subscript) = Control Group; **CDT** = Cool Detection Threshold; **CPT** = Cold Pain Threshold; **DN4** = Douleur Neuropathique 4; **DSIS** = Daily Sleep Interference Scale; **HADS** = Hospital Anxiety and Depression Scale; **HPT** = Heat Pain Threshold; **I-GRP** (subscript) = Intervention Group; **IENFD** = Intraepidermal Nerve Fiber Density; **IPQ** = Inguinal Pain Questionnaire; **Md** = Median; **Mn** = Mean; **MDT** = Mechanical Detection Threshold; **MDP** = Median Duration of Pain, all patients; **MPT** = Mechanical Pain Threshold; **MPV** = Mean Pain Values; **NP** = Neuropathic Pain; **NPQ** = Neuropathic Pain Questionnaire; **NPSI** = Neuropathic Pain Symptom Inventory; **NRS** = Numeric Rating Scale; **P-A** = Pain, Activity; **PCS** = Pain Catastrophizing Scale; **PE-I** = Psychosexual Evaluation, Interview; **P-GRP** (subscript) = Placebo Group; **POP** = Postoperative Pain; **POP-C** = Postoperative Pain, Control Group; **POP-P** = Postoperative Pain, Patient Group; **P-P** = Pain, Pressure; **PPHP** = Persistent Postherniotomy Pain; **PPT** = Pressure Pain Threshold; **P-R** = Pain, Rest; **PRIEA** = Pain Related Impairment of Everyday Activities; **QoL** = Quality of Life; **PS** (subscript) = Pain Side; **PSQI** = Pittsburgh Sleep Quality Index; **pat** = Patients; **P-WU** = Pinprick Windup; **SF-36** = Short Form 36; **SI** = Sleep Interference; **S-LANSS** = Self-reported Leeds Assessment of Neuropathic Symptoms and Signs Pain Scale; **SM-B** = Sensory Mapping, Brush; **SM-C** = Sensory Mapping, Cool; **SM-P** = Sensory Mapping, Pinprick; **S-NRS** = Summed NRS; **SPID** = Summed Pain Intensity Difference; **S-VAS-PI** = Spontaneous Visual Analog Scale Pain Intensity; **S-QBAA** = Self-reported Questionnaire Based Activity Assessment; **S-QBPA** = Self-reported Questionnaire Based Pain Assessment; **SUSP** = Swedish Universities Scale of Personality; **STHS** = Suprathreshold Heat Stimuli; **UCGP** = Unilateral Chronic Groin Pain; **WDT** = Warmth Detection Threshold, **Wk** = Week; **Yrs** = Years.

→ = No significant difference between patient group and control group

↑ = Significant increase/improvement

↓ = Significant decrease

Thermal thresholds were assessed in all 5 RCTs. A suprathreshold heat pain stimulus (STHS) was used in the assessment of pain intensity. WDT, CDT, HPT, and CPT were assessed in one RCT [42], WDT, CDT, and HPT in 1/5 studies [8], whilst the thermal paradigms in the remaining 3/5 studies [9, 29, 30] were WDT, CDT, HPT, and STHS. The thermal examinations in all RCTs were performed using a Modular Sensory Analyzer (MSA). The active thermode area was 12.5 cm$^2$ and the temperature ramp rate ± 1°C/s (Table 1).

The PPT was also measured in all RCTs using a pressure algometer, with a cut-off limit of 350 kPa (Table 1).

Assessment of temporal summation to brush and punctate stimulation was included in 1/5 RCTs [42].

Pain intensity assessments related to the QST were reported with the VAS (Visual Analog Scale) 0–10 in all studies.

Control-site examinations were performed in the contralateral groin in all 5 studies.

**3.3.2 Non-randomized controlled trials.** Sensory mapping, testing mechanical allodynia to brush, was performed in 13/20 of the studies [7, 12, 13, 15, 18, 24, 25, 27, 28, 31, 36–38], and allodynia was demonstrated in 10/20 studies [7, 12, 15, 18, 24, 25, 27, 31, 36, 37]. Cool sensitivity to thermo-roller was tested in 9/20 studies [7, 12, 15, 24, 25, 27, 36–38], and hypo-/hyperalgesia to punctate stimulation in 12/20 studies [7, 12, 13, 15, 18, 24, 25, 27, 28, 31, 36, 37].

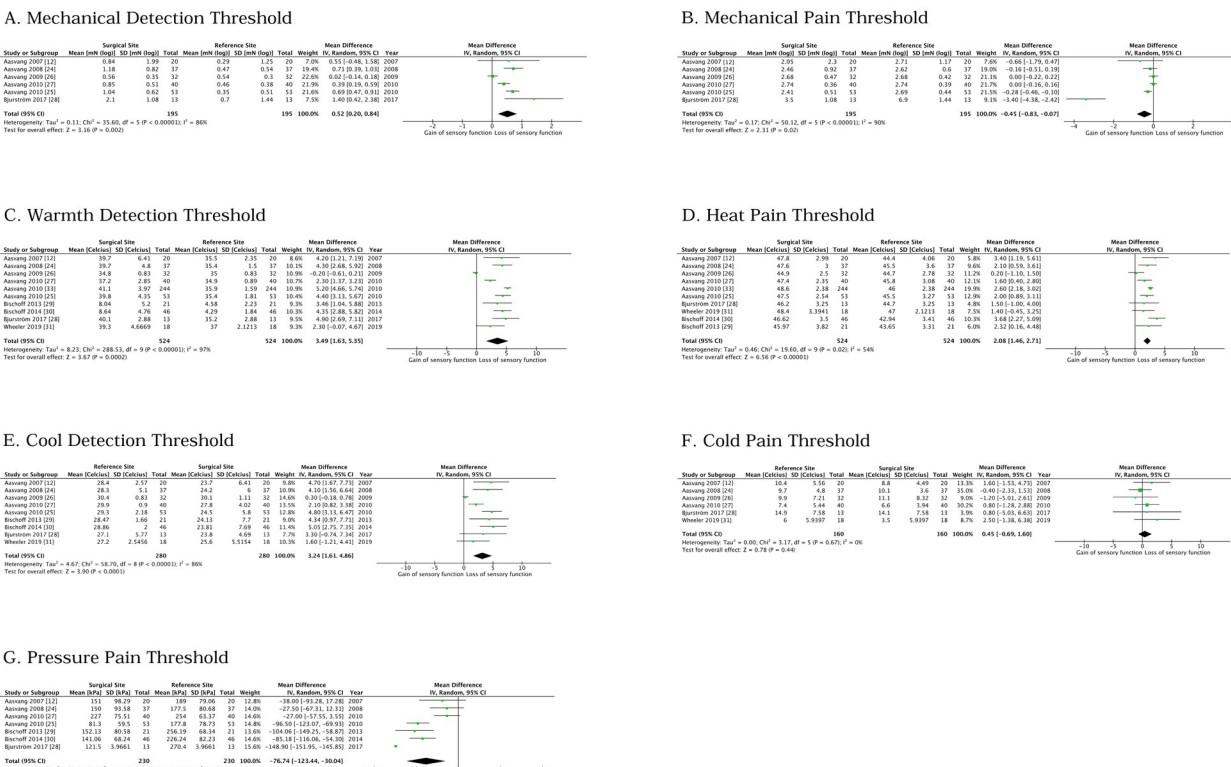

**Fig 2. Forest plots illustrating the results of various QST-modalities in the surgical site of GHR patients with PPSP compared to a reference site (contralateral side or healthy subjects).** The panels (**A-G**) illustrate assessments of mechanical detection thresholds (**A**), mechanical pain thresholds (**B**), warmth detection thresholds (**C**), heat pain thresholds (**D**), cool detection thresholds (**E**), cold pain thresholds (**F**), and blunt pressure pain thresholds (**G**) [23].

Mechanical stimuli are classified either as punctate or 'blunt' corresponding to small stimulation areas ($< 0.05$ mm$^2$) and large stimulation areas ($> 0.15$ cm$^2$), respectively [43]. As such, when referring to MDT in this review, it is associated with punctate stimulation and PPT with blunt pressure.

The MDT was examined in 15/20 studies [7, 10, 12, 13, 15, 18, 24–28, 34, 38, 39] and MPT in 14/20 studies [7, 10, 12, 13, 15, 18, 24–28, 34, 38, 39]. The MDT/MPT was evaluated using 17 progressively rigid polyamide monofilaments ranging in bending force from 0.1 to 2,941 mN in 10/20 studies [7, 10, 12, 15, 18, 24–27, 38], in one study [34] polyamide monofilaments ranging from 0.5 to 468 mN were utilized, whilst 3 studies [13, 28, 39] used monofilaments ranging from 0.7 to 2,941 mN.

Thermal thresholds were assessed in 16/20 non-RCTs [7, 10, 12, 13, 15, 18, 24–28, 31, 33, 38, 39, 44]. The thermal threshold modalities reported in 14/20 studies [7, 10, 12, 13, 15, 25–28, 31, 33, 38, 39, 44] were WDT, CDT, HPT, and CPT. In one study [18], the thermal thresholds included were WDT, CDT, and HPT, and in one study [33], WDT, HPT, and STHS were obtained. In 12/20 studies [7, 10, 12, 15, 18, 24–27, 33, 37, 38], an active thermode area of 12.5 cm$^2$ was used, while in 4/20 [13, 28, 31, 39], an active thermode area of 9 cm$^2$ was used.

The PPT was assessed in 15/20 studies [7, 10, 12, 13, 15, 18, 24–28, 35, 37–39]. In 11/20 [7, 10, 12, 15, 18, 24–27, 35, 38], the cut-off limit was set at 350 kPa, 2/20 [13, 28] had the cut-off limit set at 588 kPa, while 2/20 [37, 39] did not report any cut-off limit.

**Table 2. Surgical technique and demographic characteristics.**

| First author, publication year, ref# | Surgical method | Age (yrs) | BMI (kg/m$^2$) |
|---|---|---|---|
| **Randomized Controlled Trials** | | | |
| Aasvang EK, 2010 [42] | Open | Mn$_{I-GRP}$: 50.7 (n = 20). Mn$_{C-GRP}$: 51.4 (n = 21) | NR |
| Bischoff JM, 2012 [8] | NR | Md$_{L1\%}$ (RNG): 56 (45–71). Md$_{L2\%}$ (RNG): 44 (42–54). Md$_{L1-2\%}$ (RNG): 48 (42–67) | NR |
| Bischoff JM, 2013 [29] | Open (n = 16) vs LAP (n = 5) | Mn (SD): 57 (13) | Mn (SD): 25 (2) |
| Bischoff JM, 2014 [30] | Open (n = 38) vs LAP (n = 8) | Mn$_{I-GRP}$ (SD): 52 (17), Mn$_{P-GRP}$ (SD): 55 (14) | Mn$_{I-GRP}$ (IQR): 25 (23–30). Mn$_{C-GRP}$ (IQR): 26 (23–28) |
| Wijaysinghe N, 2016 [9] | Open | Mn$_{PAT-GRP}$ (RNG): 53 (24–83). Mn$_{hv}$ (RNG): 28 (21–41) | Mn$_{PAT-GRP}$ (SD): 25.3 (2.5). Mn$_{hv}$ (SD): 23.2 (2.1) |
| **Non-Randomized Controlled Trials** | | | |
| Mikkelsen T, 2004 [18] | Open | Md (RNG): 64 (48–73) | NR |
| Aasvang EK, 2007 [12] | Open | RNG$_{PAT-GRP}$: 20–42 | NR |
| Aasvang EK, 2008 [24] | Open | Mn$_{PAT-GRP}$ (RNG): 46 (23–70), Mn$_{C-GRP}$ (RNG): 44.2 (22–59) | NR |
| Beldi G, 2008 [34] | Open nonmesh (Grp A), Open mesh (Grp B), LAP (Grp C) | Md$_A$ (RNG): 61 (28–77), Md$_C$ (RNG): 62 (45–83), Md$_C$ (RNG): 57 (28–70) | NR |
| Aasvang EK, 2009 [26] | NR | Md (RNG): 50 (18–64) | NR |
| Aasvang EK, 2009 [15] | Open | Md (RNG): 48 (27–70) | NR |
| Kalliomäki ML, 2009 [36] | Control group: Shouldice (n = 11), Lichtenstein (n = 58), LAP (n = 14), Other (n = 9). Pain group: Shouldice (n = 8), Lichtenstein (n = 65), LAP (n = 11), Other (n = 8) | Md$_{PAT-GRP}$ (RNG): 58 (21–85), Md$_{C-GRP}$ (RNG): 59 (21–85) | NR |
| Aasvang EK, 2010 [27] | Open | Md (RNG): 56 (22–70) | NR |
| Aasvang EK, 2010 [33] | Open (n = 244) vs LAP (n = 198) | Mn (SD): 55.2 (13.3) | Mn (SD): 25,1 (2.8) |
| Aasvang EK, 2010 [25] | NR | Md$_{PAT-GRP}$ (RNG): 47 (23–76), Md$_{C-GRP}$ (RNG): 56 (22–70) | NR |
| Linderoth G, 2011 [38] | LAP | Md (RNG): 43 (23–60) | NR |
| Kristensen AD, 2012 [37] | NR | Mn (SD): 7.8 (2.6) | NR |
| van den Broeke EN, 2013 [40] | Control group: Lichtenstein (n = 2), Open nonmesh (n = 3), Open mesh (n = 2). Pain group: Open nonmesh (n = 3), Open mesh (n = 5) | Mn$_{PAT-GRP}$ (SD): 53 (11), Mn$_{C-GRP}$ (SD): 59 (7) | NR |
| Moore AM, 2016 [39] | Laparoscopic preperitoneal repair (n = 26), Open anterior (Lichtenstein) (n = 13), Plug and patch (n = 8), Open preperitoneal mesh (n = 4), Plug (n = 2), Tissue repair: Shouldice (n = 2), Tissue repair: Bassini (n = 2) | Mn (RNG): 47 (22–79) | NR |
| Bjurström MF, 2017 [28] | Open, unilateral approach (n = 6), laparoscopic, bilateral procedure (n = 4) | Mn (RNG): 46.5 (22–61) | Mn (RNG): 28.8 (24.1–38.5) |
| Bjurström MF, 2017 [13] | Open, unilateral (n = 8), open, bilateral (n = 1), laparoscopic, bilateral procedure (n = 4) | Mn (RNG): 48.8 (22–79) | Mn (RNG): 28.0 (23.2–38.5) |
| Ergönenc T, 2017 [35] | Open | Mn (SD): 57 (16.7) | Mn (SD): 28.76 (3.71) |
| Jensen EK, 2019 [7] | Open (n = 160/204), mesh-and-plug (2/204), Onstep (2/204), LAP (37/204), undecided, due to repeat surgeries (3/204) | Md (RNG): 50.1 (49.0–53.8) | NR |

(*Continued*)

**Table 2.** (Continued)

| First author, publication year, ref# | Surgical method | Age (yrs) | BMI (kg/m²) |
|---|---|---|---|
| Wheeler DW, 2019 [31] | Open | Mn (SD): 56.4 (13.5) | NR |
| Jensen EK, 2021 [10] | Open vs LAP | Md (RNG): 47.8 (45.4–50.3) | NR |

**C-GRP** (subscript) = Control Group; **I-GRP** (subscript) = Intervention Group; **IQR** = Interquartile Range; **LAP** = Laparoscopic; **Mn** = Mean; **Md** = Median; **NR** = Not Reported; **P-GRP** (subscript) = Placebo Group; **PAT-GRP** (subscript) = Patient Group; **RNG** = Range; **SD** = Standard Deviation; **L1%** (subscript) = Lidocaine, 1% Group; **L1-2%** (subscript) = Lidocaine, 1–2% Group.

Temporal summation to brush and punctate stimulation was examined in 10/20 studies [7, 10, 12, 15, 18, 24–27, 38], while temporal summation to punctate stimulation only was tested in 2/20 [13, 28] studies.

For the QST-related pain intensity assessments, 10/20 studies [7, 10, 12, 15, 24, 26, 33, 35, 38, 39] utilized NRS 0–10, 4/20 [13, 28, 34, 40] used VAS 0–10, 3/20 [18, 25, 27] assessed pain with VAS 0–100, while 2/20 studies [31, 37] did not report pain intensity assessments.

## 3.4 Study results

**3.4.1 Mechanical detection and pain thresholds (MDT/MPT).** MDT was included as a QST modality in 15 studies (1 RCT, 14 non-RCTs) [7, 10, 12, 13, 15, 18, 24–28, 34, 38, 39, 42]. Six studies [12, 24–28] were found useable for subsequent data analysis (n = 195; Fig 2A). MDT was examined on the surgical side, and data were compared with the contralateral groin. The total mean MDT difference (back-transformed) in patients with PPSP was 3.3 (95% CI = 1.6, 6.9) mN when comparing the affected side with the opposite groin. This indicates a significantly lower sensitivity, or loss of sensory function, for mechanical detection in PPSP-patients (P = 0.002). Test for heterogeneity, however, indicated a high level of heterogeneity ($I^2$ = 86%, P < 0.00001).

In the 15 studies examining MDT, there was no significant difference in MDTs between the surgical side *vs*. non-surgical side or control group in 6 studies [12, 15, 18, 25, 26, 42]. Four studies [13, 24, 27, 28] reported a significantly increased MDT in the surgical groin compared to the contralateral side. Five studies [7, 10, 34, 38, 39] provided insufficient reporting of data regarding comparison of baseline MDT-values on the surgical side, contralateral groin, or other reference data (Table 3).

MPT was also part of the QST protocol in the 15 studies. Data from 6 studies [12, 24–28] were pooled for analysis (n = 195; Fig 2B). The total mean MPT difference was -2.8 mN (95% CI = -6.8, -1.2, back-transformed), indicating near-significantly increased hyperalgesia (gain of sensory function) for punctate mechanical stimulation (P = 0.02). There was a high level of heterogeneity ($I^2$ = 90%, P < 0.00001).

In 6 of the studies [12, 15, 18, 24, 27, 42], comparison of MPT in the surgical groin vs the contralateral side showed no significant difference. Two studies [13, 28] reported a significant increase in MPT-values. The MPT on the surgical side was significantly decreased in 2 studies [25, 26], and in 5 studies [7, 10, 34, 38, 39] the outcome was insufficiently reported (Table 3).

**3.4.2 Thermal assessments (WDT, CDT, HPT, CPT, STHS).** Thermal thresholds were assessed in 21/25 studies [7–10, 12, 13, 15, 18, 24–31, 33, 37–39, 42] (5 RCTs/16 non-RCTs). WDT and HPT were measured in all 21 studies, CDT in 20/21 studies [7–10, 12, 13, 15, 18, 24–31, 37–39, 42], CPT in 16/21 studies [7, 10, 12, 13, 15, 24–28, 31, 37–39, 42] (1 RCT/15

**Table 3. QST characteristics.**

| First author, publication year, ref# | QST-related outcomes | Assessment tool/manufacturer | Pain ratings | QST-related results | Comparison (reference site/control group) | Data presentation |
|---|---|---|---|---|---|---|
| **Randomized Controlled Trials** | | | | | | |
| Aasvang EK, 2010 [42] | SM-C, SM-B, SM-P, MDT, MPT, WDT, CDT, HPT, CPT, PPT, P-WU, B-WU | SM-B: Cotton gauze SM-C: TR (20°C)/Somedic AB, SM-P: vF (0.1–2941.2 mN)/Stoelting Co. MDT, MPT: vF (0.1–2941.2 mN)/Stoelting Co. TT: MSA (TA 12.5 cm$^2$, RR +/- 1°C/s, BLT 32°C, COL 52°C & 7°C)/Somedic AB. PPT: PA (COL 350 kPa)/Somedic AB, 1 cm$^2$. WU-B: Cotton gauze, WU-P: vF (588.2 mN)/Stoelting Co. | NRS (0–10) | Pre-op: MDT→, MPT→, WDT→, CDT→, HPT→, CPT→, PPT→, P-WU→, B-WU→, 1 wk: MDT$_I$↑, 2.5 yrs: MDT→, MPT→, WDT→, CDT→, HPT→, CPT→, PPT→, P-WU→, B-WU→ | Contralateral + Control group | SM: Incidence, MDT, MPT, TT, PPT: NR (no numerical data, presented in figures), P-WU, B-WU: Incidence |
| Bischoff JM, 2012 [8] | SM-C, WDT, CDT, HPT, PPT, STHS | SM-C: TR (25°C)/Somedic AB. TT: MSA (TA 12.5 cm$^2$, RR +/- 1°C/s, BLT 32°C, COL 50°C & 5°C, STHS assessed with phasic 5-sec 47°C stimulus)/Somedic AB, 1 cm$^2$. PPT: PA (COL 350 kPa)/Somedic AB. | NRS (0–10) | WDT→, CDT→, HPT→, PPT→, STHS$_{L2\%}$↓ | Contralateral + control group | SM: Incidence (fraction of pt), TT: Mn (IQR), PPT: Mn (IQR) |
| Bischoff JM, 2013 [29] | SM-C, WDT, CDT, HPT, PPT | SM-C: TR (25°C)/Somedic AB. TT: MSA (TA 12.5 cm$^2$, RR +/- 1°C/s, BLT 32°C, COL 50°C & 5°C, STHS assessed with phasic 5-sec 47°C stimulus)/Somedic AB. PPT: PA (COL 350 kPa)/Somedic AB, 1 cm$^2$. | NRS (0–10) | WDT→, CDT→, HPT→, PPT$_{I-GRP}$↑ | Contralateral + control group | SM: Incidence (fraction of pt), TT: Mn (SD), PPT: Mn (SD) |
| Bischoff JM, 2014 [30] | SM-C, WDT, CDT, HPT, PPT, STHS | SM-C: TR (25°C)/Somedic AB. TT: MSA (TA 12.5 cm$^2$, RR +/- 1°C/s, BLT 32°C, COL 50°C & 5°C, STHS assessed with phasic 5-sec 47°C stimulus)/Somedic AB, felt-tip, 1 cm$^2$. PPT: PA (COL 350 kPa)/Somedic AB. | NRS (0–10) | WDT→, CDT→, HPT→, PPT→, STHS→ | Contralateral + control group | TT: Mn (SD), PPT: Mn (SD) |
| Wijaysinghe N, 2016 [9] | SM-C, WDT, CDT, HPT, PPT, STHS | SM-C: TR (25°C)/Somedic AB. TT: MSA (TA 12.5 cm$^2$, RR +/- 1°C/s, BLT 32°C, COL 50°C & 5°C, STHS assessed with phasic 5-sec 47°C stimulus)/Somedic AB. PPT: PA (COL 350 kPa)/Somedic AB, felt-tip, 1 cm$^2$. | NRS (0–10) | WDT→, CDT$_i$↑, HPT→, PPT$_{I-GRP}$↑, STHS$_{I-GRP}$↑ | Contralateral + control group | TT: Md (95% CI) PPT: Md (95% CI) |
| **Non-Randomized Controlled Trials** | | | | | | |
| Mikkelsen T, 2004 [18] | SM-B, SM-P, MDT, MPT, WDT, CDT, HPT, PPT, WU-P, WU-B | SM-B: Brush, SM-P: vF (0.1–468 mN)/Somedic AB. MDT, MPT: vF (0.078–468 mN)/Somedic AB. TT: MSA (TA 12.5 cm$^2$, RR +/- 1°C/s, BLT 32°C, COL 50°C & 25°C)/Somedic AB. PPT: PA (COL 350 kPa)/Somedic AB, 0.18 cm$^2$. WU-P: vF (0.078–468 mN) WU-B: ETB (Oral-B 4713, Braun) | VAS (0–100) | MDT→, MPT→, WDT→, CDT→, HPT→, STHS→, PPT→, P-WU$_{P-GRP}$↑, B-WU→ | Control group | SM: Incidence, MPT: Md (IQR), TT: Md (IQR), PPT: Md (IQR), WU: Md (IQR) |

*(Continued)*

**Table 3.** (Continued)

| First author, publication year, ref# | QST-related outcomes | Assessment tool/manufacturer | Pain ratings | QST-related results | Comparison (reference site/ control group) | Data presentation |
|---|---|---|---|---|---|---|
| Aasvang EK, 2007 [12] | SM-C, SM-B, SM-P, MDT, MPT, WDT, CDT, HPT, CPT, PPT, WU-P, WU-B | SM-C: TR (25 ˚C)/Somedic AB, SM-B: Cotton gauze, SM-P: vF (588.2 mN)/Stoelting Co., MDT/MPT: vF (0.1–2941.2 mN)/Stoelting Co. TT: MSA (TA 12.5 cm$^2$, RR +/- 1 ˚C/s, BLT 32 ˚C, COL 52 ˚C & 10 ˚C)/Somedic AB. PPT: PA (COL 350 kPa)/Somedic AB, Neoprene tip, 0.18 cm$^2$, WU-P: vF (588.2 nM)/Stoelting Co. WU-B: Cotton gauze | NRS (0–10) | MDT→, MPT→, WDT→, CDT→, HPT→, CPT→, PPT→, WU$_{P-GRP}$: 8/10 experienced pain, WU$_c$: 11/20 experienced pain | Contralateral + control group | SM: Incidence, MDT: Mn (95% CI) MPT: Mn (95% CI), TT: Md (IQR), PPT: Md (IQR), WU: Incidence |
| Aasvang EK, 2008 [24] | SM-C, SM-B, SM-P, MDT, MPT, WDT, CDT, HPT, CPT, PPT, WU-P, WU-B | SM-C: TR (20 ˚C)/Somedic AB, SM-B: Cotton gauze, SM-P: vF (588.2 mN)/Stoelting Co. MDT/ MPT: vF (0.1–2941.2 mN)/ Stoelting Co. TT: MSA (TA 12.5 cm$^2$, RR +/- 1 oC/s, BLT 32 ˚C, COL 52 ˚C & 10 ˚C)/Somedic AB. PPT: PA (COL 350 kPa)/ Somedic AB, Neoprene tip, 0.18 cm$^2$, WU-P: vF (588.2 nM)/ Stoelting Co. WU-B: Cotton gauze | NRS (0–10) | WDT$_p$↑, CDTp↑, HPT→, CPT→, MDT$_p$↑, MPT→, PPT$_p$↓, WU-P$_{P-GRP}$: 30/37 experienced pain, WU-B$_{P-GRP}$: 7/37 experienced pain | Contralateral + control group | SM: Incidence, MDT: Mn (95% CI) MPT: Mn (95% CI), TT: Md (IQR), PPT: Md (IQR), WU: Incidence |
| Beldi G, 2008 [34] | MDT, MPT | MDT/MPT: (0.5–468 mN)/ Somedic AB | VAS (0–10) | Insufficient reporting of data | No reference site or control group | NR |
| Aasvang EK, 2009 [26] | MDT, MPT, WDT, CDT, HPT, CPT, PPT, WU-P, WU-B | MDT/MPT: vF (0.078–2941.2 mN)/Stoelting Co., TT: MSA (TA 12.5 cm2, RR +/- 1˚C/s, BLT 32˚C, COL 52˚C & 5˚C)/ Somedic AB, PPT: PA (COL 350 kPa)/Somedic AB, Neoprene tip, 0.18 cm$^2$ WU-P: vF (588.2 nM)/Stoelting Co. WU-B: Cotton Gauze | NRS (0–10) | MDT→, MPT$_p$↓, WDT→, CDT→, HPT→, CPT→, PPT→, WU$_{OP-P}$: 3/42 experienced pain, WU-B→ | Contralateral side, no control group | MDT: Mn (95% CI) MPT: Mn (95% CI), TT: Mn (95% CI), PPT: Mn (95% CI) |
| Aasvang EK, 2009 [15] | SM-B, SM-C, SM-P, MDT, MPT, WDT, CDT, HPT, CPT, PPT, WU-P, WU-B | SM-B: Cotton gauze, SM-C: TR (20˚C)/Somedic AB, SM-P: vF (588.2 mN)/Stoelting Co., MDT/MPT: vF (0.1–2941.2 mN)/Stoelting Co. TT: MSA (TA 12.5 cm2, RR +/- 1˚C/s, BLT 32˚C, COL 52˚C & 5˚C)/ Somedic AB, PPT: PA (COL 350 kPa)/Somedic AB, Neoprene tip, 0.18 cm$^2$ WU-P: vF (588.2 nM)/Stoelting Co. WU-B: Cotton gauze | NRS (0–10) | MDT→, MPT→, WDT$_{OP}$↑, CDT→ HPT$_{OP}$↑, CPT→, PPT$_{OP}$↑, WU-P$_{op}$: 9/21 experienced pain, WU-B→ | Contralateral side, no control group | NR |
| Kalliomäki ML, 2009 [36] | SM-B, SM-C, SM-P | SM-B: Brush/SENSELab, SM-C: TR (25˚C)/SENSELab, SM-H: TR (40˚C)/SENSELab, SM-P: NR | NRS (0–10) | Insufficient reporting of data | Control group | NR |

(Continued)

**Table 3.** (Continued)

| First author, publication year, ref# | QST-related outcomes | Assessment tool/manufacturer | Pain ratings | QST-related results | Comparison (reference site/control group) | Data presentation |
|---|---|---|---|---|---|---|
| Aasvang EK, 2010 [27] | SM-B, SM-C, SM-P, MDT, MPT, WDT, CDT, HPT, CPT, PPT, WU-P, WU-B | SM-B: Cotton gauze, SM-C: TR (20°C)/Somedic AB, SM-P: vF (588.2 mN)/Stoelting Co., MDT/MPT: vF (0.08–2941.2 mN)/Stoelting Co. TT: MSA (TA 12.5 cm², RR +/- 1°C/s, BLT 32°C, COL 52°C & 7°C)/ Somedic AB, PPT: PA (COL 350 kPa)/Somedic AB, Neoprene tip, 0.18 cm², WU-P: vF (588.2 nM)/Stoelting Co. WU-B: Cotton gauze | VAS (0–100) | MDT$_{OP}$↑, MPT→, WDT$_{OP}$↑, CDT$_{OP}$↑, HPT$_{OP}$↑, CPT→, PPT→, WU-P$_{OP}$: 6/40 experienced pain, WU-P$_{CL}$: 2/40, WU-B$_{OP}$: 1/40 experienced pain | Contralateral side, no control group | MDT: Mn (95% CI) MPT: Mn (95% CI), TT: Mn (95% CI), PPT: Mn (95% CI), WU: Incidens |
| Aasvang EK, 2010 [33] | WDT, HPT, STHS | TT: MSA (TA 12.5 cm², RR +/- 1°C/s, BLT 32°C, COL 52°C & 7°C)/Somedic AB | NRS (0–10) | WDT$_{OP-OS}$↑, WDT$_{OP-LS}$↓, HPT$_{OP-OS}$↑, HPT$_{OP-LS}$↑, STHS→ | Arm as reference site, no control group | TT: Mn (95% CI) |
| Aasvang EK, 2010 [25] | SM-B, SM-C, SM-P, MDT, MPT, WDT, CDT, HPT, CPT, PPT, WU-P, WU-B | SM-B: Cotton gauze, SM-C: TR (20°C)/Somedic AB, SM-P: vF (588.2 mN)/Stoelting Co., MDT/MPT: vF (0.1–2941.2 mN)/Stoelting Co. TT: MSA (TA 12.5 cm², RR +/- 1°C/s, BLT 32°C, COL 52°C & 5°C)/ Somedic AB, PPT: PA (COL 350 kPa)/Somedic AB, neoprene tip, 1 cm², WU-P: vF (588.2 nM)/Stoelting Co. WU-B: Cotton gauze | VAS (0–100) | MDT→, MPT$_{P-OP}$↓, WDT$_{P-OP}$↑, CDT$_{P-OP}$↑, HPT$_{P-OP}$↑, CPT→, PPT$_{P-OP}$↓, WU-P$_{P-GRP}$: 36/70 experienced pain, WU-B$_{P-GRP}$: 15/70 experienced pain | Contralateral, control group | MDT: Mn (95% CI) MPT: Mn (95% CI), TT: Mn (95% CI), PPT: Md (IQR), WU: Incidens |
| Linderoth G, 2011 [38] | SM-C, MDT, MPT, WDT, CDT, HPT, CPT, PPT, WU-P, WU-B | SM-C: TR (25°C)/Somedic AB, MDT/MPT: vF (0.1–2941.2 mN)/Stoelting Co. TT: MSA (TA 12.5 cm², RR +/- 1°C/s, BLT 32°C, COL 50°C & 5°C)/ Somedic AB, PPT: PA (COL 350 kPa)/Somedic AB, 1 cm², WU-P: vF, WU-B: Br | NRS (0–10) | Insufficient reporting of data | Contralateral side, no control group | NR (no numerical data) |
| Kristensen AD, 2012 [37] | SM-B, SM-C, SM-P, WDT, CDT, HPT, CPT, PPT | SM-B: Brush (Brush-05)/ SENSELab, SM-C: TR (20°C)/ SENSELab, SM-P: vF (588.2 nM)/Stoelting Co, TT: MSA (TA 12.5 cm², RR +/- 1°C/s, BLT 32°C, COL 50°C & 10°C)/ Somedic AB, PPT: PA/Somedic AB, 1 cm². | NR | WDT→, CDT→, HPT→, CPT→, PPT$_{P-OP}$↓ | Contralateral side, no control group | NR (insufficient data, only 3 pt) |
| van den Broeke EN, 2013 [40] | Hypoesthesia (brush, monofilament) + evoked or increased pain (brush) | NR | VAS (0–10) | Insufficient reporting of data | Contralateral + control group | NR |
| Moore AM, 2016 [39] | MDT, MPT, WDT, CDT, HPT, CPT, PPT | MDT/MPT: vF (0.7–2942 mN)/ North Coast Medical., TT: TSA-II (TA 9 cm², RR +/- 1°C/ s, BLT 32°C, COL 50°C & 0°C)/ Medoc Ltd, PPT: PA/Wagner Instruments, 1 cm² | NRS (0–10) | Insufficient reporting of data | Contralateral side, no control group | NR (no numerical data) |

(Continued)

**Table 3.** (Continued)

| First author, publication year, ref# | QST-related outcomes | Assessment tool/manufacturer | Pain ratings | QST-related results | Comparison (reference site/control group) | Data presentation |
|---|---|---|---|---|---|---|
| Bjurström MF, 2017 [28] | SM-P, MDT, MPT, WDT, CDT, HPT, CPT, PPT, WU-P, WU-B | SM-P: PC (PA 0.25 mm$^2$) MDT/MPT: vF (0.7–2942 mN)/North Coast Medical., TT: TSA-II (TA 9 cm$^2$, RR +/- 1˚C/s, BLT 32˚C, COL 50˚C & 0˚C)/Medoc Ltd, PPT: PA (COL 588 kPa)/Wagner Instruments, 1 cm$^2$. WU-P: vF (98 nM)/North Coast Medical | VAS (0–10) | MDT$_{I-GRP}$↑, MPT$_{I-GRP}$↑, WDT$_{I-GRP}$↑, CDT$_{I-GRP}$↑, HPT$_{I-GRP}$↑, CPT$_{I-GRP}$↑, PPT$_{I-GRP}$↑, WU-P: No pat. experienced pain after intervention, WU-B: No pt experienced pain after intervention | Contralateral side, no control group | MDT: Mn (SE) MPT: Mn (SE), TT: Mn (SE), PPT: Mn (SE) |
| Bjurström MF, 2017 [13] | SM-P, MDT, MPT, WDT, CDT, HPT, CPT, PPT, WU-P, WU-B | SM-P: PC (PA 0.25 mm$^2$) MDT/MPT: vF (0.7–2942 mN)/North Coast Medical., TT: TSA-II (TA 9 cm2, RR +/- 1˚C/s, BLT 32˚C, COL 50˚C & 0˚C)/Medoc Ltd, PPT: PA (COL 588 kPa)/Wagner Instruments, 1 cm$^2$. WU-P: vF (98 nM)/North Coast Medical | VAS (0–10) | MDT$_{I-GRP}$↑, MPT$_{I-GRP}$↑, WDT$_{I-GRP}$↑, CDT$_{I-GRP}$↑, HPT$_{I-GRP}$↑, CPT$_{I-GRP}$↑, PPT$_{I-GRP}$↑, WU-P: No pt experienced pain after intervention WU-B: No pt experienced pain after intervention | Contralateral side, no control group | MDT: Mn (SE) MPT: Mn (SE), TT: Mn (SE), PPT: Mn (SE) |
| Ergönenc T, 2017 [35] | PPT | PPT: Dolorimeter, 1 cm$^2$ | NRS (0–10) | PPT$_{P-OP}$↓ | Contralateral side, control group | PPT: Mn (SD) NP: Presented as kg/cm2, not kPa |
| Jensen EK, 2019 [7] | SM-B, SM-C, SM-P, MDT, MPT, WDT, CDT, HPT, CPT, PPT, WU-P, WU-B | SM-B: Cotton gauze, SM-C: TR (22˚C)/Somedic AB, SM-P: vF (588.2 mN)/Stoelting Co., MDT/MPT: vF (0.1–2941.2 mN)/Stoelting Co. TT: MSA (TA 12.5 cm$^2$, RR +/- 1˚C/s, BLT 32˚C, COL 52˚C & 5˚C, STHS assessed with phasic 5-sec 47˚C stimulus)/Somedic AB, PPT: PA (COL 350 kPa)/Somedic AB, neoprene tip, 1 cm$^2$, WU-P: vF (588.2 nM)/Stoelting Co. WU-B: Br | NRS (0–10) | NR (QST data not presented) | Contralateral side + arm as reference sites, no control group | NR |
| Wheeler DW, 2019 [31] | SM-B, SM-P, WDT, CDT, HPT, CPT | SM-P: vF (255 nM)/Semmes-Weinstein, SM-B: Cotton Q-tip/Unilever, TT: Pathway ATS (TA 30 mm$^2$, RR +/- 1˚C/s, BLT 32˚C, COL 51˚C/Medoc | NR | WDT$_{OP}$↑, CDT→, HPT$_{OP}$↑, CPT$_{OP}$↑ | Contralateral side, no control group | WDT: Mn (SD), CDT: Mn (SD), HPT: Mn (SD) |
| Jensen EK, 2021 [10] | MDT, MPT, WDT, CDT, HPT, CPT, PPT, WU-P, WU-B | MDT/MPT: vF (0.1–2941.2 mN)/Stoelting Co. TT: MSA (TA 12.5 cm2, RR +/- 1˚C/s, BLT 32˚C, COL 52˚C & 5˚C, SHPT assessed with phasic 5-sec 47˚C stimulus)/Somedic AB, PPT: PA (COL 350 kPa)/Somedic AB, neoprene tip, 1 cm$^2$, WU-P: vF (588.2 nM)/Stoelting Co. WU-B: Brush | NRS (0–10) | NR (QST data not presented) | Contralateral side + arm as reference sites, no control group | NR |

**L2%** (subscript) = Control Group, 2% Lidocaine **BLT** = Baseline Temperature; **CDT** = Cool Detection Threshold; **CoL** = Cutoff Limit; **CPT** = Cold Pain Threshold; **HPT** = Heat Pain Threshold; **I-GRP** (subscript) = Intervention Group; **MDT** = Mechanical Detection Threshold; **MPT** = Mechanical Pain Threshold; **NRS** = Numeric Rating Scale; **OP-AAS** (subscript) = Operated Side, Patients With Increased AAS-score; **OP-LS** (subscript) = Operated Side, Laparoscopic Surgery; **OP-OS** (subscript) = Operated Side, Open Surgery; **P-GRP** (subscript) = Placebo Group; **PA** = Pressure Algometer; **PPT** = Pressure Pain Threshold; **RR** = Ramp Rate; **SM-B** = Sensory Mapping, Brush; **SM-C** = Sensory Mapping, Cool; **SM-P** = Sensory Mapping, Pinprick; **STHS** = Suprathreshold Heat Stimulus; **TA** = Thermode Area; **TR** = Thermoroll; **TT** = Thermal Thresholds; **vF** = von Frey Monofilament; **WDT** = Warmth Detection Threshold, **WU-B** = Windup-Brush; **WU-P** = Windup-Pinprick

→ = No significant difference between patient group and control group

↑ = Significant increase/improvement

↓ = Significant decrease

non-RCTs) and STHS in 4 studies [8, 9, 18, 30, 33] (3 RCTs/1 non-RCT). In one study [33], an area of the arm was used as reference site, while the contralateral groin served as reference area in the remaining studies (Table 3).

For the meta-analysis, data for WDT were pooled from 11 studies [12, 24–27, 29–31, 33] (2 RCTs/9 non-RCTs; n = 722) with the contralateral side as a control (Fig 2C). The total mean difference was 3.2 (95% CI = 1.6, 4.7) ˚C indicating a loss of sensory function on the surgical side (P = 0.0001). However, a high level of heterogeneity was seen ($I^2$ = 97%, P = 0.00001).

Out of the 21 studies examining WDT, 8 studies (all non-RCTs) [13, 15, 24, 25, 27, 28, 31, 33] found significant increases in the WDT on the surgical site of PPSP-patients following GHR. However, 9 studies [8, 9, 12, 18, 29, 30, 37, 42] reported no significant difference in WDT between sides. Interestingly, in the study by Aasvang et al. [33], those who underwent laparoscopic surgery experienced a significant *decrease* in WDT, in contrast to the cohort operated by an open technique. The remaining 4 studies [7, 10, 38, 39] reported insufficient data for analysis of WDT (Table 3).

Outcome data for HPT were pooled from 10 studies [12, 24–27, 29–31, 33] (n = 722; Fig 2D). The mean difference in HPT was 1.9 (95% CI = 1.1, 2.7) ˚C, indicating a significant increase in the surgical, painful groin compared with a reference area. Again, a high level of heterogeneity was present ($I^2$ = 82%, P = 0.00001). In total, 7 studies [13, 15, 25, 27, 28, 31, 33] found a significant increase in HPT in the painful, surgical groin corresponding to a sensory loss of function. In 10 studies [8, 9, 12, 18, 24, 26, 29, 30, 37, 42], the investigators did not find a significant difference in HPT, and in 4 studies [7, 10, 38, 39], insufficient reporting of data was seen (Table 3).

Outcome data regarding CDT were pooled from 9 studies [12, 24–31] (n = 280; Fig 2E). The mean difference in CDT was 3.2 (95% CI = 1.6, 4.9) ˚C, indicating a significant loss of sensory function (P = 0.0001). The level of heterogeneity was high ($I^2$ = 86%, P = 0.00001). Six of the studies [9, 13, 24, 25, 27, 28] which corroborated a significant increase in CDT when comparing the surgical side with the contralateral side, while 10 studies [8, 12, 15, 18, 26, 29–31, 37, 42] found no significant changes in CDT. In 4 of the studies [7, 10, 38, 39] there was insufficient reporting of CDT outcome data (Table 3).

For the analysis of CPT, data from 6 studies [12, 24, 26–28, 31] were pooled (n = 160; Fig 2F). Compared to the control area, there were no significant differences in CPTs (mean difference 0.5 (95% CI = -0.7, 1.6) ˚C (P = 0.44)). Heterogeneity was low ($I^2$ = 0, P = 0.67). In total, 3 studies [13, 28, 31] found a significant numerical increase in CPT in the surgical groin, compared to the control area. No significant differences were found in the remaining studies (Table 3).

STHS was applied in 4 studies [8, 9, 30, 33] (3 RCTs, 1 non-RCTs). A comprehensive analysis was not possible due to the format of data presentation in the studies. One study [29] found a statistically significant decreased STHS-induced pain intensity in the control group receiving 2% lidocaine blockade in the painful site. In the study by Wijaysinghe et al. [9], the intervention group had a significantly decreased STHS-induced pain intensity after bupivacaine tender point blockade. The remaining 2 studies [8, 30] examining STHS did not find a significant difference in induced pain intensities when comparing the painful site with a reference area or a control group (Table 3).

**3.4.3 Pressure pain threshold (PPT).** PPT was assessed in 20 studies [7–10, 12, 13, 15, 18, 24–30, 35, 37–39, 42] (5 RCTs/15 non-RCTs). Data were pooled from 7 studies [12, 24, 25, 27–30] (n = 230). The mean difference in the analysis of PPT was -76 (95% CI = -123, -30) kPa, indicating a significant gain of sensory function (increased pain sensitivity) in the surgical groin compared to a reference area or control group (P = 0.001). The level of heterogeneity was high ($I^2$ = 96%, P < 0.00001) (Fig 2G).

In 2 of the RCTs [9, 29], significant PPT increases in the surgical site were found following intervention with 5% lidocaine patch or 0.25% bupivacaine tender point block. There was no significant difference in PPT in the remaining RCTs [8, 30, 42]. For the non-RCTs, 4 studies [12, 18, 26, 27] found no significant difference in PPT. In 4 of the studies [24, 25, 35, 37], there was a significant decrease in PPT, whilst 3 studies found a significant increase [13, 15, 28]. The remaining 4 non-RCTs [7, 10, 38, 39] showed insufficient reporting related to the outcome (Table 3).

### 3.5 Quality assessment

**3.5.1 Randomized controlled trials (RoB 2.0).** An overview of the assessments is provided in Fig 3. Three of the 5 studies were deemed "low" risk of bias [8, 9, 29]. The study by Aasvang et al [42] showed "some concerns" in domain 5 (bias in selection of the reported result), resulting in the overall judgement of "some concerns" of risk of bias. The study by Bischoff et al. [30] was assessed as having "high" risk of bias in domain 4 (bias in the measurement of the outcome), resulting in an overall judgment of "high" risk of bias. The assessment was justified by the fact that the blinding of investigators and study subjects would be problematic due to the pungent smell and stinging sensory properties of the capsaicin patch compared to the inert placebo patch.

**3.5.2 Non-randomized controlled trials (Newcastle Ottawa Scale).** For the cross-sectional studies, a modified version of the NOS was used [45]. A summary of the NOS bias assessments is provided in Table 4. A full overview of the quality gradings is available as S1 Table. Cross-sectional studies and S2 Table. Cohort studies. The NOS bias assessments summarized:

Only 3 of the cohort studies [13, 28, 39] failed to meet the criteria related to "representativeness of the exposed cohort", as the cohorts consisted of highly selected study subjects. Additionally, 3 of the cohort studies [31, 33, 39] provided a "demonstration that outcome of interest was not present at the start of the study" [21]. Not a single study was allocated a "star" under "assessment of the outcome" since every study either used self-reported questionnaires/interviews or because the outcomes related to QST were, in fact, self-reported by the study participant. Further comments regarding the NOS are provided in the discussion section below.

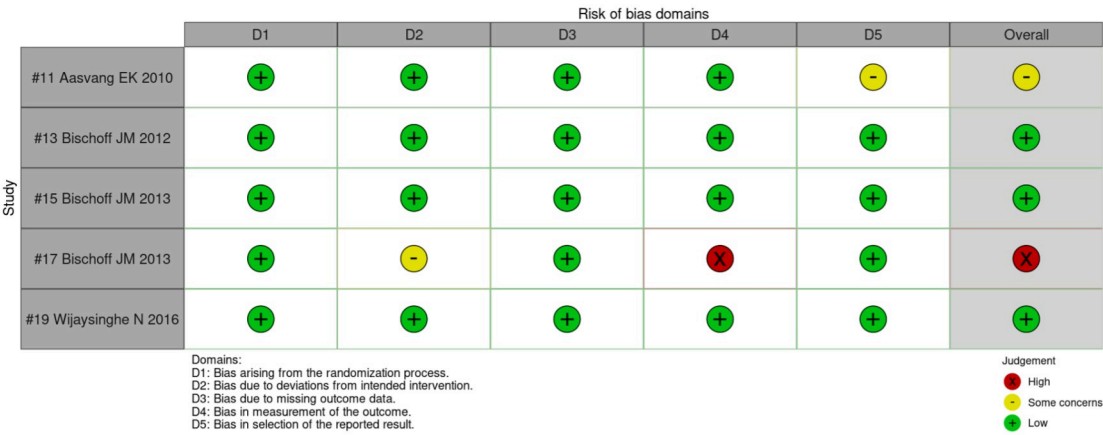

**Fig 3. Summary of the risk of bias assessments of included RCTs using RoB 2.0 [22].**

**Table 4. Summary of NOS bias assessments.**

| Cross-sectional studies | | | | |
|---|---|---|---|---|
| Refs. | Author | Year | Title | Total score (Maximum 9 stars) |
| 18 | Mikkelsen | 2004 | Pain and sensory dysfunction 6 to 12 months after inguinal herniotomy | 7 |
| 26 | Aasvang | 2009 | Preoperative pain and sensory function in groin hernia | 6 |
| 27 | Aasvang | 2010 | Persistent sensory dysfunction in pain-free herniotomy | 8 |
| 37 | Kristensen | 2012 | Chronic pain after inguinal hernia repair in children | 4 |
| 35 | Ergönenç | 2017 | Persistent postherniorrhaphy pain following inguinal hernia repair | 7 |
| Cohort studies | | | | |
| Refs. | Author | Year | Title | Total score (Max 9) |
| 12 | Aasvang | 2007 | Ejaculatory pain: a specific postherniotomy pain syndrome? | 6 |
| 24 | Aasvang | 2008 | Neurophysiological characterization of postherniotomy pain | 7 |
| 34 | Beldi | 2008 | Postoperative hypoesthesia and pain: qualitative assessment after open and laparoscopic inguinal hernia repair | 7 |
| 15 | Aasvang | 2009 | The effect of mesh removal and selective neurectomy on persistent postherniotomy pain | 7 |
| 36 | Kalliomäki | 2009 | Persistent pain after groin hernia surgery: a qualitative analysis of pain | 7 |
| 33 | Aasvang | 2010 | Predictive risk factors for persistent postherniotomy pain | 8 |
| 25 | Aasvang | 2010 | Heterogeneous sensory processing in persistent postherniotomy pain | 6 |
| 38 | Linderoth | 2011 | Neurophysiological characterization of persistent pain after laparoscopic inguinal hernia repair | 7 |
| 40 | van den Broeke | 2013 | Altered cortical responsiveness to pain stimuli after high frequency electrical stimulation of the skin in patients with persistent pain after inguinal hernia repair | 7 |
| 39 | Moore | 2016 | Efficacy of retroperitoneal triple neurectomy for refractory neuropathic inguinodynia (Prospective study) | 7 |
| 28 | Bjurström | 2017 | Neurophysiological and clinical effects of laparoscopic retroperitoneal triple neurectomy in patients with refractory postherniorrhaphy neuropathic inguinodynia | 6 |
| 13 | Bjurström | 2017 | Quantitative validation of sensory mapping in persistent postherniorrhaphy inguinal pain patients undergoing triple neurectomy | 6 |
| 7 | Jensen | 2019 | A national center for persistent severe pain after groin hernia repair: Five-year prospective data | 7 |
| 31 | Wheeler | 2019 | Evaluation of postsurgical hyperalgesia and sensitization after open inguinal hernia repair: A useful model for neuropathic Pain? | 8 |
| 10 | Jensen | 2021 | Trajectories in severe persistent pain after groin hernia repair: a retrospective analysis | 7 |

## 4. Discussion

### 4.1 Short summary

The aims of this systematic review were, *first*, to identify and describe the available literature on the use of QST in patients with PPSP following GHR, and, *second*, to explore the role of QST in understanding mechanisms underlying PPSP following GHR. To the best of our knowledge, this is the first review to systematically assess the use of somatosensory testing in this cohort. The review, based on 25 studies (5 RCTs, 20 non-RCTs), delivers, a qualitative synthesis of the findings coupled with a meta-analysis of data obtained across eligible studies.

In 15/25 studies, mechanical assessments, MDT, and MPT were assessed. Thermal assessments WDT, CDT, HPT, CPT, or STHS, were included as QST-modalities in 21/25 studies and PPT in 20/25 studies. Analysis of pooled data from GHR patients with PPSP showed significant differences between the surgical side and reference sites in MDT and MPT (n = 195; Fig 2A and 2B); WDT (n = 722; Fig 2C); HPT (n = 722, Fig 2D); CDT (n = 280, Fig 2E) and PPT (n = 230, Fig 2G). However, no significant difference in CPT was found (n = 160, Fig 2F).

## 4.2 Methodological quality assessment

**4.2.1 Cochrane Risk-of-Bias Tool 2.0.**   As mentioned, the RoB 2.0 was used for the methodological quality assessment of included RCTs and NOS for non-RCTs. For the quality assessments of the RCTs with RoB 2.0, we found the tool intuitive to use, and the Cochrane Collaboration has provided thorough guidance for the application of the tool. This resulted in less doubt when conducting the methodological quality assessments avoiding ambiguous interpretations of how to use the tool. It should be noted that RoB 2.0 is the gold standard for quality assessments of RCTs, whilst no such standard is set for non-RCTs as to the best of our knowledge.

**4.2.2 Newcastle-Ottawa Scale.**   Although the NOS is a validated tool for assessment of risk of bias in non-RCT studies, concerns have previously been raised regarding the use of this instrument [46, 47]. While a manual for the tool is provided [21], we found the available guidance on the application of the tool to be somewhat sparse. This lack of clear guidance and our limited experience with the tool led to repeated complex discussions regarding the obtained results between the present authors.

## 4.3 Mechanical assessments

**4.3.1 Punctate mechanical thresholds.**   *Methodological issues*. Methodological concerns regarding the use of monofilaments for assessment of MDT and MPT, include regular calibration. Polyamide monofilaments are affected by the relative humidity [43, 48] and usage [49]. The calibration procedure entails measurement of bending force for each monofilament using a precision weight and recording of the relative humidity of the environment by an electronic hygrometer. Notably, one percent increase in relative humidity corresponds to a 1–4% relative decrease in numerical bending force, depending on the diameter of the monofilament [50]. Seasonal variations in relative humidity are common, meaning that the bending force may fluctuate over the course of a study. Calibration curves across different relative humidities have been published [43].

In the included studies assessing MDT and MPT, the monofilaments used were handheld by the investigator, which could result in a variability in impact angle and contact area between the monofilament and skin [50–52].

Only five studies [13, 26, 28, 34, 39] specifically stated that the monofilaments were calibrated. A single study [18] mentioned that the monofilaments were calibrated at a specific relative humidity and temperature (35%, 23˚C). In the included studies assessing MDT and MPT, 8/15 [13, 15, 25–28, 33, 42] mentioned that the QST was performed in a room with a temperature of 20–24˚C. However, the bending force is not affected within a temperature range of 22–30˚C [43].

*Outcomes—Mechanical detection threshold*. Based on our analyses (Fig 2A), MDT was significantly increased (P = 0.002) on the surgical site in patients with PPSP compared to the contralateral groin. As such, patients affected by PPSP following GHR experience hypoesthesia for punctate stimulation (loss of sensory function). A single study has been conducted to establish normative data on sensory function in pain-free post-herniotomy patients [27]. The study included 40 patients, who had all undergone open surgery. A significant increase (hypoesthesia) in MDT, was demonstrated when comparing the surgical side with the contralateral side, corroborating our findings.

*Outcomes—Mechanical pain threshold*. Regarding MPT, our analyses indicated a near-significant difference in the surgical site compared to the contralateral groin (P = 0.02; Fig 2B). Two studies [13, 25] showed a significant decrease in MPT (gain of sensory function), with one of them [13] being an outlier when compared to the other data. Five out of 13 patients

included in the study had previously undergone remedial operations without meaningful improvement. The surgical interventions included "groin re-exploration, replacement or mesh-removal and unsuccessful neurectomy of the ilioinguinal and iliohypogastric nerves". Likely, this would indicate that the 5/13 patients experienced advanced neuropathic pain associated with mechanical hyperalgesia [13]. This could explain the deviations, the gain of sensory function as well as the increase in data variance seen in this outlying cohort. Interestingly, prior to the triple neurectomy performed in the study, the patients had a VAS-score ranging from 5 at best to 9.5 at worst. Six months after the triple neurectomy, the VAS-scores were 1.5 and 5.5 respectively.

**4.3.2 Blunt pressure pain threshold.** *Methodological issues.* All 20 studies that included PPT as a QST-modality used a handheld pressure algometer for assessment of blunt pressure (Table 3). Several studies have demonstrated high reliability when assessing PPT with hand-held pressure algometers in various anatomical regions [53–56].

However, differences across studies, related to the probes' contact areas could contribute to data variability (heterogeneity). Six of the 15 studies [12, 15, 18, 24, 26, 27] assessing PPT used a probe with a surface area of 0.18 cm$^2$, while the remaining 9 studies used a contact area of 1 cm$^2$. It is important to consider the stimulation area, since differences in probe area size could influence results due to differences in stimulation of superficial and deeper situated nociceptors (skin *vs.* fascia) [1, 57]. If the same force was to be applied with a 0.18 cm$^2$ probe and a 1 cm$^2$ probe, the pressure would be reduced by a factor 5.6. This would result in a significant indentation when using the larger probe compared to the smaller probe.

As with the monofilaments, calibration can likewise potentially affect the equipment used for assessing PPT. This is however a mere speculation—no study has been conducted with the purpose of comparing calibrated vs non-calibrated handheld pressure algometers. None of the included studies mentioned whether the pressure algometers used were properly calibrated prior to assessments.

*Outcomes.* As a single modality, blunt PPT was most frequently included as part of the QST protocols in the studies. In total, 9/20 studies found significant differences in PPT. Five of these studies were intervention studies investigating the effects of a lidocaine patch [29], ultrasound-guided tender point blockade [9], and triple neurectomy [13, 15, 28]. Based on the analysis, PPT was significantly decreased in the surgical site compared to the contralateral groin.

## 4.4 Thermal assessments

**4.4.1 Methodological issues.** An important aspect of thermal assessments is the active thermode area used for assessing thermal thresholds, i.e., spatial summation. Most of the studies used a rectangular thermode with a surface area of 12.5 cm$^2$ (Table 3), while three studies [13, 28, 39] used a quadratic size of 9 cm$^2$. When comparing thermal thresholds considerable deviations may occur if differing active thermal areas are used uncritically, as shown in the study by Rasmussen et al. [58]. Most noticeably, one study [31] stated the use of a thermode size of 30 mm$^2$, while in the supplementary files for the study, an area of 9 cm$^2$ is mentioned.

None of the included studies commented on regular calibration procedures of the equipment used for thermal assessments.

**4.4.2 Detection thresholds.** Our analyses of thermal detection thresholds showed significant increases in WDT and absolute values of CDT, indicating a loss of small fiber sensory function (cf. section 4.5.).

**4.4.3 Pain thresholds.** Our analyses of thermal pain thresholds showed a significant increase in HPT, indicating a loss of small fiber sensory function, while no significant differences were found regarding CPT (cf. section 4.5.).

**4.4.4 Outcome: Suprathreshold heat stimulus.** Pain assessments with suprathreshold heat stimuli (STHS) were also included in 5 studies [8, 9, 18, 30, 33], with various results. As shown in Table 3, some studies found significant increases in pain intensities induced by STHS, which could indicate an involvement of central sensitization in patients with PPSP following GHR [10, 59].

## 4.5 Cutaneous *vs.* deep mechanical thresholds

The surgical procedure and the type of mesh implant could also have an impact on the transition to chronic pain. Minimally invasive repair seems to be linked to a lower incidence of postoperative complications, e.g., hematoma, wound infection, a lower prevalence of persistent pain, and an earlier return to work/daily activities [1]. Further research is, however, needed to minimize confounding variables obscuring the results [1, 2]. In addition to groin hernia repair, other surgical procedures such as breast implants, vascular grafts, and joint prosthetic material are also known to be associated with pathophysiologic events related to mesh implants [3]. Iatrogenic nerve damage and the gradual onset of neuropathic pain may be brought on by surgical dissection or transection of the nerve or by fixation of the mesh (sutures, tacks). Additionally, the mesh implant is prone to dehiscence, dislocation, induration, invasion of nearby structures, or shrinkage, processes that may result in a 20–90% reduction in mesh area [4, 5].

Studies have shown a loss of intraepidermal nerve fiber density (IENFD) on the surgical side in groin hernia repair patients when compared with the contralateral, healthy groin [29, 30]. This decrease in IENFD could serve as an explanation for the hyposensitivity ("loss of function") in WDT, CDT, HPT, and MDT. However, no significant differences in sensitivities were found for CPT and MPT, in spite of a loss of small fiber sensory function (theoretically resulting in increased thresholds). This paradoxical finding indicates the presence of a compensatory central sensitization phenomenon.

In the case of hyperalgesia ("gain of function") for blunt pressure stimulation, the issue likely resides in the deeper tissues. When assessing PPT, the pressure is applied on the point of maximum palpatory evoked pain, an area that relates to the superficial inguinal ring. The opening in the abdominal wall is associated with several important anatomical structures, including the vas deferens, vascular supply to the testicles, the ilioinguinal nerve, and the genital branch of the genitofemoral nerve [7]. When performing blunt pressure algometry, pressure is applied to the superficial ring, compressing these deeper structures, including part of the implanted mesh [1]. A severely inflamed mesh may develop into a pathological "meshoma" [6]. Histologically, a "meshoma" is a granulomatous process that mechanically or by inflammation may affect or compress adjacent tissues, e.g., the spermatic cord or nerves, developing into a "pain generator". Furthermore, peripheral nerves such as the ilioinguinal nerve and the genital branch of the genitofemoral nerve have been demonstrated to become embedded in mesh material leading to pain by mechanical and inflammatory reactions [1].

Two of the studies included in our review have specifically addressed the "pain generator" [8, 9]. The studies performed in patients with severe persistent pain following groin hernia repair were double-blind, crossover RCTs applying an ultrasound-guided blockade. In the first study (n = 12), the iliohypogastric and ilioinguinal nerves were targeted at the level of the anterior superior iliac spine [8]. However, the blockades had no effect on the PPSP, possibly indicating that a block of the genitofemoral nerve instead was necessary to achieve pain reduction. In the second study (n = 14), blockades at the tender point located above the superficial inguinal ring were examined [9]. A median decrease in pain was observed, i.e., 63% compared to 36% after placebo (P = 0.003) [9]. Although the pain relief was found to be short lasting, the results suggested that peripheral afferent input from the tender point area has an essential role

in the preservation of evoked and spontaneous pain in PPSP following groin hernia repair. In addition, several studies have found that a surgical approach, e.g., meshectomy, or, selective or triple neurectomy, may result in a significant reduction of pain compared to control groups [15, 60]. Furthermore, a recently published study [61] indicates that re-surgery in the form of meshectomy and selective neurectomy provides increases in thermal and punctate mechanical thresholds. The study importantly reports highly significant increases in PPT following re-surgery supporting that meshectomy and selective neurectomy may slow the "pain generator".

As such, the pathophysiology behind PPSP in GHR-patients could partially be explained by partial deafferentation of cutaneous nerve fibers in combination with the development of a "pain generator" in deeper layers, e.g., subepidermal structures and fascia layers. Re-innervation and neo-innervation in mesh implants and in indigenous tissue are well-known phenomena in herniorrhaphy patients. Interestingly, in patients where pain is the reason for mesh-excision, the mesh neural innervation has been shown to be significantly higher in comparison to patients where the mesh was excised because of recurrence [4].

## 4.6 Limitations

**4.6.1 Studies.** *Number of studies*. One limitation of the review is the limited number of eligible studies. Whereas our comprehensive search strategy identified 25 studies, we were only able to pool data from 11 of the studies for quantitative analysis.

*Level of heterogeneity*. The studies differed regarding applied QST modalities, methodological quality, statistical processing, and data presentation. However, a caveat is that in most of the studies, QST variables were not part of the main outcome, which could explain the observed heterogeneity across studies. Although some of the studies cited the DFNS paradigm, comprising seven tests with 11 stimulation modalities and the assessment of 13 somatosensory variables, the complete paradigm was not used in any of the studies [14]. Test durations of $27 \pm 2.3$ min per test area have been reported in healthy volunteers [14]. Applying the complete testing paradigm on individuals in severe pain at two to three locations may cause individual distress and fatigue, and potentially affect the reliability of somatosensory testing. Definitions of "moderate" and "severe" pain also differed between studies, with discrepancies in the NRS score (0–10) corresponding to a particular intensity of pain. For example, some studies defined severe pain as NRS ranging from 8–10 [11, 26], whilst others defined it as NRS $\geq$ 6 [29, 42] or $\geq$ 7 [38] (Table 1). Standardized definitions of pain intensities, as proposed by Collins et al [62], could be beneficial in reducing uncertainties in the literature regarding pain patients.

*Test-retest reliability*. Assessing the reliability of QST data is important for correct interpretation. The sensory perturbations caused by surgery are highly variable between individuals but also within individuals [61]. Therefore, addressing the variability by test-retest analyses is necessary for evaluating validity of QST data. However, one study [63] reported test-retest reliability using secondary study data from healthy volunteers. Interestingly, thus, it does not seem that test-retest data are currently available in patients with persistent pain after groin hernia repair.

*Healthy controls vs. contralateral side*. The controlled studies used different methodological approaches when comparing the sensory abnormalities at the surgical site to a control site. Either an *absolute* approach, comparing with a normative healthy cohort, or a *relative* approach, comparing with the individual's contralateral homotopic site, or a combination of these approaches, were used. One of the advantages of the *relative* approach is that the within-subject variances often are significantly smaller than the between-subject variances. Using the individual's contralateral site as a control is thus expected to reduce data variability, making the data more robust and less susceptible to confounding factors such as age,

gender, and random errors. On the other hand, mirror-image sensory dysfunction [64], a neural cross-talk between the sides, may influence the side-to-side difference in the relative approach. Very few studies have systematically examined the pros and cons of the absolute and relative approaches [65].

**4.6.2 Review methodology.** *Systematic vs. narrative approach.* The limited number of studies available for quantitative analyses, 11/25, infers that a narrative analytical approach was necessary for the remaining studies.

*Meta-analytical approach.* Due to the limited data accessibility in the pooled analyses and the large general heterogeneity of the studies, the authors decided to designate the examination as a "meta-analytical approach". Nevertheless, the forest plots provide relevant and meaningful patterns of postsurgical sensory dysfunction (Fig 2A–2G).

## 5. Conclusions

This systematic review critically examined all published literature related to quantitative somatosensory testing in patients with persistent pain after groin hernia repair. Twenty-five studies were included; significant heterogeneity regarding methodology, outcome assessment, data synthesis, risk of bias, and overall quality was encountered. Based on a meta-analytical approach applied to 11/25 studies, quantitative analyses indicated significant sensory perturbations on the operated side, i.e., loss of sensory function regarding cutaneous thresholds and a gain of sensory function regarding deep tissue stimulation. These results indicate that hyperalgesia originating from deeper tissues is a potential key element in development of persistent pain after groin hernia repair. Cutaneous deafferentation may contribute to hyperalgesia either directly or indirectly by central sensitization.

## Supporting information

**S1 Checklist. Prisma 2020 checklist.**
(DOCX)

**S1 File. Search string.**
(DOCX)

**S1 Table. Table 4—NOS (Cross-sectional studies).**
(DOCX)

**S2 Table. Table 5—NOS (Cohort studies).**
(DOCX)

## Acknowledgments

We gratefully acknowledge the assistance of research and study librarian Annemette Møller Hansen M.Sc., Copenhagen University Library in refining the search strategy.

## Author Contributions

**Conceptualization:** Akhmedkhan Dubayev, Mads U. Werner.

**Data curation:** Akhmedkhan Dubayev, Mads U. Werner.

**Formal analysis:** Akhmedkhan Dubayev, Elisabeth Kjær Jensen, Kenneth Geving Andersen, Martin F. Bjurström, Mads U. Werner.

**Investigation:** Akhmedkhan Dubayev, Elisabeth Kjær Jensen, Martin F. Bjurström, Mads U. Werner.

**Methodology:** Akhmedkhan Dubayev, Elisabeth Kjær Jensen, Martin F. Bjurström, Mads U. Werner.

**Project administration:** Akhmedkhan Dubayev, Mads U. Werner.

**Supervision:** Kenneth Geving Andersen, Martin F. Bjurström, Mads U. Werner.

**Validation:** Akhmedkhan Dubayev, Elisabeth Kjær Jensen, Mads U. Werner.

**Visualization:** Akhmedkhan Dubayev, Mads U. Werner.

**Writing – original draft:** Akhmedkhan Dubayev, Mads U. Werner.

**Writing – review & editing:** Akhmedkhan Dubayev, Elisabeth Kjær Jensen, Kenneth Geving Andersen, Martin F. Bjurström, Mads U. Werner.

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
