## [Decision Letter · Decision Letter 0]

23 Jun 2023

PONE-D-23-14833Quantitative somatosensory assessments in persistent pain after groin hernia repair: A systematic review with a meta-analytical approachPLOS ONE

Dear Dr. Dubayev,

Thank you for submitting your manuscript to PLOS ONE. After careful consideration, we feel that it has merit but does not fully meet PLOS ONE’s publication criteria as it currently stands. Therefore, we invite you to submit a revised version of the manuscript that addresses the points raised during the review process.

We look forward to receiving your revised manuscript.

Kind regards,

Armando Almeida

Academic Editor

PLOS ONE

Journal Requirements:

Additional Editor Comments:

REVIEWER 1:

I congratulate the authors for their work of thoroughly revising the literature in this field. I believe that this is a relevant study to help characterize the somatosensory alterations that may underlie persistent pain after GHR.

However, I have some comments and suggestions that I believe would help improve the quality, comprehensibility and utility of this work. From the reader’s perspective, I was left with some doubts.

One general comment is that the timing of QST assessment should be made clearer in the title and in the abstract (as it is in section 1.1.3). There are many studies that assess QST preoperatively and then use that assessment to longitudinally predict pain chronification. In this review, it the authors only consider QST assessment made after the surgery, when patients already have criteria for diagnosis of PPSP. I think that it should be clearly stated that this is a review of QST in patients with PPSP. This is not entirely transparent in the title.

The inclusion criteria should also be clearer. It is not clear if any QST test was eligible for inclusion (as stated in the abstract: “For inclusion, studies had to report at

80 least one QST-modality in patients with PPSP”), or if only thermal and mechanical modalities were considered.

Also, the discussion section could be improved, by providing more information concerning potential pathophysiological mechanisms explaining the sensory deficits found in GHR patients. This would be important to meet aim nº 2 “examine the role of QST in outlining the pathophysiological mechanisms behind PPSP after GHR.”

Please, find my detailed comments below.

Introduction

First paragraph: I suggest that the authors revise this paragraph, since they seem to go back and forth in the ideas presented. For example, the sentence in line 128 would be more coherent earlier in the paragraph (maybe after ref. 4,5?). Also, some ideas are repeated, for example lines 129-130 and line 123.

Page 6, line 123: Could the authors specify which these more elaborate criteria were? If they are alluded to, they should be summarized in the text. Otherwise, the sentence is not informative.

Page 6, line 136: I believe that the authors present three causes, not two: inflammation, meshoma ot neuropathy.

Page 7, line 147: As I mentioned previously, I suggest that this sentence should be revised for clarity. During my first read, I wondered if the authors meant “use of QST in PSPS” or “use of QST to predict PPSP”.

Page 7, line 148: This sentence is not clear: “In relation to the treatment of PPSP, the surgical intervention, either in terms of meshectomy, selective or triple neurectomy, is quite effective in a subset of patients”. Do the authors mean that patients undergo a second surgery to treat PPSP?

Page 8, line 171: If this review analyzes QST in patients already with PSPP, how it is expected to contribute to prediction and prognostic outcomes?

Materials and Methods

Page 9, line 191: QST assesses more than just detection thresholds and pain thresholds of mechanical and thermal stimuli (for example, pain summation and electrical stimuli). Why was this criteria adopted? If the inclusion criteria only included “detection thresholds and pain thresholds for mechanical and thermal stimuli”, the aims should be revised (“retrieve and methodologically characterize the available literature related to QST in patients with PPSP following GHR”).

Other questions concerning inclusion criteria are:

- Were studies evaluating conditioned pain modulation also excluded?

- What was the minimum time-period after surgery for studies to be included? Three months?

- What age groups where considered for inclusion? Adults only?

Page 9, line 198: At least a summary of search terms should be present in the main manuscript.

Page 10, line 225: The authors state that “In non-randomized clinical trials (non-RCTs), the Newcastle Ottawa Scale (NOS) [21] and the Cochrane Risk of Bias Tool 2.0 (RoB 2.0) [22], were used”. However,the Cochrane tool was used for RCT, as stated below in the text (line 237).

Page 11, line 245: The main objective of what? The review? The data synthesis? Either way, this sentence does not seem accurate or needed in this section

Page 12, line 264: Why did the authors choose to only present log data for MDT and MPT? I would say that data presentation should be homogenous.

Page 12, line 274: Please reconsider the word “arbitrarily”. If there was a specific intention behind the choice of this value, it does not seem to have been arbitrary.

Results

Page 13, line 290-293: In this section, it is described that twenty studies where non-RCTs and five of those were cross-sectional. What about the other 15?

Page 28, line 386: Allodynia would be better described as mechanical hypersensitivity.

Page 30, line 229: What are these values?: (1.6, 6.9)

Page 42, line 529-532: In the methods section, the authors mention that for RCTs, only pretreatment QST will be considered. However, in this paragraph, they report decreases in pain in the groups receiving treatment. Wouldn’t it be more relevant to present pre-treatment comparisons? The aim here is not to present QST changes after treatment. The same comment applies for results concerning PPT.

Page 43, line 570: There should be a summary table of NOS bias assessment in the main manuscript.

Page 44, line 580: The first sentence is too long. Please consider rewrite. Also, I would suggest replacing the term “study subjects” by “patients”.

Page 44, lines 596-611: The considerations made here concerning RoB are not adequate for a scientific discussion.

Page 46, line 643: Please consider if it would be more correct to state “near-significant difference” instead of “change”. There is no change being assessed, only a side-to-side comparison.

Page 46, line 651-655: Can the authors explain why this reduction in pain intensity is worth mentioning?

Page 47, line 680: Can the authors further explore the meaning of these results? Otherwise, this section is a simple statement of results. The same applies to the discussion of suprathreshold heat stimulus.

Page 48, line 681: There is no discussion of results for WDT, CDT, HPT and CPT.

Throughout the discussion, I would like to have found more information concerning the second goal “to examine the role of QST in outlining the putative pathophysiological mechanisms behind PPSP”. I think that the potential mechanisms could be more thoroughly approached.

Also, do the authors believe that the surgical approach could also influence QST the results? Are there different percentages of PPSP according to surgical approach? I think that this variable should be focus of discussion, since it is a relevant source of heterogeneity.

REVIEWER 2:

This is an important study to characterize persistent pain after groin hernia repair (GHR). Importantly, no systematic/methodological review of the findings on the use of QST in PPSP following GHR has been published until now. Care was taken to perform the best methodologies and scales throughout the study. I have only a few points to be better elucidated.

Comment #1: Page 28/58 – “MDT and MPT were included as QST-modalities in one RCT using 17 progressively rigid polyamide monofilaments ranging in bending force from 0.1 to 2941 mN” – were all of them von Frey filaments? Did their use for getting mechanical thresholds follow the up-and-down method by Chaplan et al 1994? Please explain better.

Comment #2: A high level of heterogeneity was present in several evaluations. How might this interfere with a good evaluation of the results?

Reviewers' comments:

Reviewer's Responses to Questions

**Comments to the Author**

1. Is the manuscript technically sound, and do the data support the conclusions?

Reviewer #1: Yes

Reviewer #2: Yes

2. Has the statistical analysis been performed appropriately and rigorously? 

Reviewer #1: Yes

Reviewer #2: Yes

3. Have the authors made all data underlying the findings in their manuscript fully available?

Reviewer #1: Yes

Reviewer #2: Yes

4. Is the manuscript presented in an intelligible fashion and written in standard English?

Reviewer #1: Yes

Reviewer #2: Yes

5. Review Comments to the Author

Reviewer #1: To the authors:

I congratulate the authors for their work of thoroughly revising the literature in this field. I believe that this is a relevant study to help characterize the somatosensory alterations that may underlie persistent pain after GHR.

However, I have some comments and suggestions that I believe would help improve the quality, comprehensibility and utility of this work. From the reader’s perspective, I was left with some doubts.

One general comment is that the timing of QST assessment should be made clearer in the title and in the abstract (as it is in section 1.1.3). There are many studies that assess QST preoperatively and then use that assessment to longitudinally predict pain chronification. In this review, it the authors only consider QST assessment made after the surgery, when patients already have criteria for diagnosis of PPSP. I think that it should be clearly stated that this is a review of QST in patients with PPSP. This is not entirely transparent in the title.

The inclusion criteria should also be clearer. It is not clear if any QST test was eligible for inclusion (as stated in the abstract: “For inclusion, studies had to report at

80 least one QST-modality in patients with PPSP”), or if only thermal and mechanical modalities were considered.

Also, the discussion section could be improved, by providing more information concerning potential pathophysiological mechanisms explaining the sensory deficits found in GHR patients. This would be important to meet aim nº 2 “examine the role of QST in outlining the pathophysiological mechanisms behind PPSP after GHR.”

Please, find my detailed comments below.

Introduction

First paragraph: I suggest that the authors revise this paragraph, since they seem to go back and forth in the ideas presented. For example, the sentence in line 128 would be more coherent earlier in the paragraph (maybe after ref. 4,5?). Also, some ideas are repeated, for example lines 129-130 and line 123.

Page 6, line 123: Could the authors specify which these more elaborate criteria were? If they are alluded to, they should be summarized in the text. Otherwise, the sentence is not informative.

Page 6, line 136: I believe that the authors present three causes, not two: inflammation, meshoma ot neuropathy.

Page 7, line 147: As I mentioned previously, I suggest that this sentence should be revised for clarity. During my first read, I wondered if the authors meant “use of QST in PSPS” or “use of QST to predict PPSP”.

Page 7, line 148: This sentence is not clear: “In relation to the treatment of PPSP, the surgical intervention, either in terms of meshectomy, selective or triple neurectomy, is quite effective in a subset of patients”. Do the authors mean that patients undergo a second surgery to treat PPSP?

Page 8, line 171: If this review analyzes QST in patients already with PSPP, how it is expected to contribute to prediction and prognostic outcomes?

Materials and Methods

Page 9, line 191: QST assesses more than just detection thresholds and pain thresholds of mechanical and thermal stimuli (for example, pain summation and electrical stimuli). Why was this criteria adopted? If the inclusion criteria only included “detection thresholds and pain thresholds for mechanical and thermal stimuli”, the aims should be revised (“retrieve and methodologically characterize the available literature related to QST in patients with PPSP following GHR”).

Other questions concerning inclusion criteria are:

- Were studies evaluating conditioned pain modulation also excluded?

- What was the minimum time-period after surgery for studies to be included? Three months?

- What age groups where considered for inclusion? Adults only?

Page 9, line 198: At least a summary of search terms should be present in the main manuscript.

Page 10, line 225: The authors state that “In non-randomized clinical trials (non-RCTs), the Newcastle Ottawa Scale (NOS) [21] and the Cochrane Risk of Bias Tool 2.0 (RoB 2.0) [22], were used”. However,the Cochrane tool was used for RCT, as stated below in the text (line 237).

Page 11, line 245: The main objective of what? The review? The data synthesis? Either way, this sentence does not seem accurate or needed in this section

Page 12, line 264: Why did the authors choose to only present log data for MDT and MPT? I would say that data presentation should be homogenous.

Page 12, line 274: Please reconsider the word “arbitrarily”. If there was a specific intention behind the choice of this value, it does not seem to have been arbitrary.

Results

Page 13, line 290-293: In this section, it is described that twenty studies where non-RCTs and five of those were cross-sectional. What about the other 15?

Page 28, line 386: Allodynia would be better described as mechanical hypersensitivity.

Page 30, line 229: What are these values?: (1.6, 6.9)

Page 42, line 529-532: In the methods section, the authors mention that for RCTs, only pretreatment QST will be considered. However, in this paragraph, they report decreases in pain in the groups receiving treatment. Wouldn’t it be more relevant to present pre-treatment comparisons? The aim here is not to present QST changes after treatment. The same comment applies for results concerning PPT.

Page 43, line 570: There should be a summary table of NOS bias assessment in the main manuscript.

Page 44, line 580: The first sentence is too long. Please consider rewrite. Also, I would suggest replacing the term “study subjects” by “patients”.

Page 44, lines 596-611: The considerations made here concerning RoB are not adequate for a scientific discussion.

Page 46, line 643: Please consider if it would be more correct to state “near-significant difference” instead of “change”. There is no change being assessed, only a side-to-side comparison.

Page 46, line 651-655: Can the authors explain why this reduction in pain intensity is worth mentioning?

Page 47, line 680: Can the authors further explore the meaning of these results? Otherwise, this section is a simple statement of results. The same applies to the discussion of suprathreshold heat stimulus.

Page 48, line 681: There is no discussion of results for WDT, CDT, HPT and CPT.

Throughout the discussion, I would like to have found more information concerning the second goal “to examine the role of QST in outlining the putative pathophysiological mechanisms behind PPSP”. I think that the potential mechanisms could be more thoroughly approached.

Also, do the authors believe that the surgical approach could also influence QST the results? Are there different percentages of PPSP according to surgical approach? I think that this variable should be focus of discussion, since it is a relevant source of heterogeneity.

Reviewer #2: This is an important study to characterize persistent pain after groin hernia repair (GHR). Importantly, no systematic/methodological review of the findings on the use of QST in PPSP following GHR has been published until now. Care was taken to perform the best methodologies and scales throughout the study. I have only a few points to be better elucidated.

Comment #1: Page 28/58 – “MDT and MPT were included as QST-modalities in one RCT using 17 progressively rigid polyamide monofilaments ranging in bending force from 0.1 to 2941 mN” – were all of them von Frey filaments? Did their use for getting mechanical thresholds follow the up-and-down method by Chaplan et al 1994? Please explain better.

Comment #2: A high level of heterogeneity was present in several evaluations. How might this interfere with a good evaluation of the results?

6. PLOS authors have the option to publish the peer review history of their article (what does this mean?). If published, this will include your full peer review and any attached files.

Reviewer #1: No

Reviewer #2: No

---

## [Author Response · Author response to Decision Letter 0]

21 Aug 2023

The following respond is also uploaded as a word-file named "Respone_to_reviewers". 

PONE-D-23-14833

Response to PLOS ONE revision letter 

Quantitative somatosensory assessments in patients with persistent pain following groin hernia repair: A systematic review with a meta-analytical approach

Dear Academic Editor Armando Almeida, 

We sincerely thank you for the decision letter and the invitation to revise our manuscript to meet the publication criteria of PLOS ONE. Furthermore, we express our gratitude for the highly relevant and thoughtful revision comments from you and the two Reviewers.

Please, find below our point-by-point replies to the Reviewers' comments. The Reviewers' comments are numbered chronologically (C1, C2, C3…) and according to the Reviewer’s # (R1, R2) followed by the authors' answers (A1, A2...) with changes indicated in red font color. Page and line numbers in this response refer to the newly revised manuscript (with track-changes, simple markup). 

Best regards on behalf of the authors,

Akhmedkhan Dubayev, M.D., and Mads U. Werner, M.D., Ph.D.

REVIEWER #1

C1R1: One general comment is that the timing of QST assessment should be made clearer in the title and in the abstract (as it is in section 1.1.3). There are many studies that assess QST preoperatively and then use that assessment to longitudinally predict pain chronification. In this review, it the authors only consider QST assessment made after the surgery, when patients already have criteria for diagnosis of PPSP. I think that it should be clearly stated that this is a review of QST in patients with PPSP. This is not entirely transparent in the title.

A1R1: We have changed the title to the following: “Quantitative somatosensory assessments in patients with persistent pain following groin hernia repair: A systematic review with a meta-analytical approach”. We hope that this minor adjustment in our title now more clearly underlines that our systematic review covers the use of QST in patients with PPSP. 

C2R1: The inclusion criteria should also be clearer. It is not clear if any QST test was eligible for inclusion (as stated in the abstract: “For inclusion, studies had to report at least one QST-modality in patients with PPSP”), or if only thermal and mechanical modalities were considered.

A2R1: We agree that our initial description of the inclusion criteria may have been unclear. We have now rephrased the sentence to (page 9, line 194): “Studies were included if at least one standard mechanical (MDT, MPT, PPT) and/or one thermal (WDT, CDT, HPT, CPT) modality related to QST were described in the study”. 

C3R1: Also, the discussion section could be improved, by providing more information concerning potential pathophysiological mechanisms explaining the sensory deficits found in GHR patients. This would be important to meet aim nº 2 “examine the role of QST in outlining the pathophysiological mechanisms behind PPSP after GHR.”

A3R1: The following changes have now been added to further describe the potential pathophysiological mechanisms behind PPSP after GHR (pages 50-52, lines 719-782): 

“4.5. Cutaneous vs. deep mechanical thresholds 

The surgical procedure and the type of mesh implant could also have an impact on the transition to chronic pain. Minimally invasive repair seems to be linked to a lower incidence of postoperative complications, e.g., hematoma, wound infection, a lower prevalence of persistent pain, and an earlier return to work/daily activities [1]. Further research is, however, needed to minimize confounding variables obscuring the results [1, 2]. In addition to groin hernia repair, other surgical procedures such as breast implants, vascular grafts, and joint prosthetic material are also known to be associated with pathophysiologic events related to mesh implants [3]. Iatrogenic nerve damage and the gradual onset of neuropathic pain may be brought on by surgical dissection or transection of the nerve or by fixation of the mesh (sutures, tacks). Additionally, the mesh implant is prone to dehiscence, dislocation, induration, invasion of nearby structures, or shrinkage, processes that may result in a 20–90% reduction in mesh area [4, 5]. 

Studies have shown a loss of intraepidermal nerve fiber density (IENFD) on the surgical side in groin hernia repair patients when compared with the contralateral, healthy groin [29, 30]. This decrease in IENFD could serve as an explanation for the hyposensitivity (“loss of function”) in WDT, CDT, HPT, and MDT. However, no significant differences in sensitivities were found for CPT and MPT, in spite of a loss of small fiber sensory function (theoretically resulting in increased thresholds). This paradoxical finding indicates the presence of a compensatory central sensitization phenomenon.

In the case of hyperalgesia (“gain of function”) for blunt pressure stimulation, the issue likely resides in the deeper tissues. When assessing PPT, the pressure is applied on the point of maximum palpatory evoked pain, an area that relates to the superficial inguinal ring. The opening in the abdominal wall is associated with several important anatomical structures, including the vas deferens, the vascular supply to the testicles, the ilioinguinal nerve, and the genital branch of the genitofemoral nerve [6]. When performing the blunt pressure algometry, pressure is applied to the skin above the superficial ring, compressing these deeper structures, including part of the implanted mesh [1]. A severely inflamed mesh may develop into a pathological ‘ʻmeshomaʼ [7]. Histologically, a ʻmeshomaʼ is a granulomatous process that mechanically or by inflammation may affect or compress adjacent tissues, e.g., the spermatic cord or nerves, developing into a “pain generator”. Furthermore, peripheral nerves such as the ilioinguinal nerve and the genital branch of the genitofemoral nerve have been demonstrated to become embedded in mesh material leading to pain by mechanical and inflammatory reactions [1]. 

Two of the studies included in our review have specifically addressed the “pain generator” [8, 9]. The studies performed in patients with severe persistent pain following groin hernia repair were double-blind, crossover RCTs applying an ultrasound-guided blockade. In the first study (n = 12), the iliohypogastric and ilioinguinal nerves were targeted at the level of the anterior superior iliac spine [8]. However, the blockades had no effect on the PPSP, possibly indicating that a block of the genitofemoral nerve instead was necessary to achieve pain reduction. In the second study (n = 14), blockades at the tender point located above the superficial inguinal ring were examined [9]. A median decrease in pain was observed, i.e., 63% compared to 36% after placebo (P = 0.003) [9]. Although the pain relief was found to be short lasting, the results suggested that peripheral afferent input from the tender point area has an essential role in the preservation of evoked and spontaneous pain in PPSP following groin hernia repair. In addition, several studies have found that a surgical approach, e.g., meshectomy, or, selective or triple neurectomy, may result in a significant reduction of pain compared to control groups [10, 11]. Furthermore, a recently published study [12] indicates that re-surgery in the form of meshectomy and selective neurectomy provides increases in thermal and punctate mechanical thresholds. The study importantly reports highly significant increases in PPT following re-surgery supporting that meshectomy and selective neurectomy may slow the “pain generator”. 

As such, the pathophysiology behind PPSP in GHR patients could partially be explained by partial deafferentation of cutaneous nerve fibers in combination with the development of a “pain generator” in deeper layers, e.g., subepidermal structures and fascia layers. Re-innervation and neo-innervation in mesh implants and in indigenous tissues are well-known phenomena in herniorrhaphy patients. Interestingly, in patients where pain is the reason for mesh-excision, the mesh neural innervation has been shown to be significantly higher in comparison to patients where the mesh was excised because of recurrence [4].” 

C4R1: First paragraph: I suggest that the authors revise this paragraph, since they seem to go back and forth in the ideas presented. For example, the sentence in line 128 would be more coherent earlier in the paragraph (maybe after ref. 4,5?). Also, some ideas are repeated, for example lines 129-130 and line 123.

A4R1: Changes to the text have now been made to limit repetitiveness and establish a more coherent paragraph (page 6, lines 117-130):

“Groin hernia repair (GHR) is a common surgery performed in more than 20 million patients worldwide every year [1]. Persistent postsurgical pain (PPSP) following GHR is a well-known medical complication [2] and efficacious management of patients with PPSP remains a major challenge for the healthcare profession [3]. The IASP (the International Association for the Study of Pain [ICD-11]) criteria for chronic post-surgical or posttraumatic pain are “a chronic pain that develops or increases in intensity after a surgical procedure or a tissue injury and persists beyond the healing process, i.e., at least three months after the surgery or tissue trauma” [4]. More elaborate criteria have previously been proposed [5]. The condition can significantly impair the physical and psychosocial functions of the individual, and a conservative estimate is that 2 % of patients undergoing groin hernia repair will be affected by PPSP [6, 7]. The prevalence of PPSP is primarily contingent on the respective surgical procedure, whilst patient-related pre-surgical factors also affect the frequency and severity of PPSP [1, 8, 9].” 

C5R1: Page 6, line 123: Could the authors specify which these more elaborate criteria were? If they are alluded to, they should be summarized in the text. Otherwise, the sentence is not informative.

A5R1: We have chosen to delete the sentence “More elaborate criteria have previously been proposed [3]”. Further, the meshoma reference has been added [7] (page 6, line 138). 

C6R1: Page 6, line 136: I believe that the authors present three causes, not two: inflammation, meshoma or neuropathy.

A6R1: Correct! Accordingly, we have amended from two to three causes: “The three primary causes of chronic pain after groin hernia repair are inflammatory processes caused by foreign materials, formation of a meshoma, and the development of neuropathic pain, e.g., nerve injury, through nerve transection, compression or entrapment or devascularization.”

C7R1: Page 7, line 147: As I mentioned previously, I suggest that this sentence should be revised for clarity. During my first read, I wondered if the authors meant “use of QST in PSPS” or “use of QST to predict PPSP”. 

A7R1: We have changed the sentence from “use of QST in PPSP following GHR” to “use of QST in patients suffering from PPSP following GHR” (page 7, line 145) to clarify that our review focuses on patients who already experience PPSP after the primary groin hernia repair, and not to predict PPSP. We hope that this change clarifies our point. 

C8R1: Page 7, line 148: This sentence is not clear: “In relation to the treatment of PPSP, the surgical intervention, either in terms of meshectomy, selective or triple neurectomy, is quite effective in a subset of patients”. Do the authors mean that patients undergo a second surgery to treat PPSP?

A8R1: Yes, indeed. The following change was implemented: “In relation to the treatment of PPSP following GHR, re-surgery, either in terms of meshectomy, or selective or triple neurectomy, is quite effective in treating a subset of patients……” (page 7, lines 146-148).

C9R1: Page 8, line 171: If this review analyzes QST in patients already with PPSP, how it is expected to contribute to prediction and prognostic outcomes?

A9R1: Thank you for the comment. The sentence has been amended: “From a pathophysiological perspective, the review may facilitate evaluation of the diagnostic efficiency of QST methods, resulting in an enhanced prediction of prognostic outcomes for PPSP and, consequentially, improved treatment paradigms” (page 8, lines 169-171). 

C10R1: Page 9, line 191: QST assesses more than just detection thresholds and pain thresholds of mechanical and thermal stimuli (for example, pain summation and electrical stimuli). Why was this criteria adopted? If the inclusion criteria only included “detection thresholds and pain thresholds for mechanical and thermal stimuli”, the aims should be revised (“retrieve and methodologically characterize the available literature related to QST in patients with PPSP following GHR”).

Other questions concerning inclusion criteria are:

- Were studies evaluating conditioned pain modulation also excluded?

- What was the minimum time-period after surgery for studies to be included? Three months?

- What age groups where considered for inclusion? Adults only?

A10R1: Thank you for the very relevant comments. Points taken. We agree that QST includes several modalities beyond mechanical and thermal stimulation, e.g., chemical, electrical, and magnetic stimulation. In addition, the stimulation patterns may involve temporal summation, spatial summation, and suprathreshold stimulation going beyond the classic detection and pain thresholds. Furthermore, QST may involve dynamic assessments (brush, roller) used in the dynamic mapping of sensory dysfunctions, e.g., primary and secondary hyper-esthesia/-algesia and allodynia. 

The Reviewer is absolutely correct. The last part of the sentence, “… with the purpose of determining detection thresholds and pain thresholds for mechanical and thermal stimuli,” is inaccurate. The sentence has been changed to (page 9, line 190-195): “There were no exclusion criteria related to surgical technique; both open and laparoscopic surgeries were considered. Studies were included if at least one standard mechanical (MDT, MPT, PPT) and/or one thermal (WDT, CDT, HPT, CPT) modality related to QST was described in the study.” 

Interestingly, the QST modalities, included in the retrieved studies predominantly considered thermal and mechanical thresholds, as seen in Tables 1 and 3. 

Studies evaluating conditioned pain modulation (CPM) were not specifically excluded. No CPM studies fulfilled the inclusion criteria.

We did not exclude studies based on the post-surgical time period. However, we do mention that for the pain to be considered “chronic, postsurgical pain”, it had to be persistent for at least three months, as stated by the ICD-11 definition (page 6, line 121-125) [13]. All of the included studies examined the respective cohorts after a post-surgical time period of at least three months. 

All age groups were included. However, only a single study assessed children [14]. This study was not included in the data analysis (Fig. 2), as only three children were assessed with QST. 

C11R1: Page 9, line 198: At least a summary of search terms should be present in the main manuscript.

A11R1: The full search string has now been included in the manuscript (page 10, lines 202-215).

C12R1: Page 10, line 225: The authors state that “In non-randomized clinical trials (non-RCTs), the Newcastle Ottawa Scale (NOS) [21] and the Cochrane Risk of Bias Tool 2.0 (RoB 2.0) [22], were used”. However, the Cochrane tool was used for RCT, as stated below in the text (line 237)

A12R1: Thank you for the observation. We have rephrased it as follows (page 11, line 246-248): “The Newcastle Ottawa Scale (NOS) [21] and the Cochrane Risk of Bias Tool 2.0 (RoB 2.0) [22] were used to evaluate the methodological quality and risk-of-bias of non-RCT’s and RCT’s, respectively.”

C13R1: Page 11, line 245: The main objective of what? The review? The data synthesis? Either way, this sentence does not seem accurate or needed in this section.

A13R1: We agree that the sentence seems redundant in this section. We have deleted the following from the paragraph: “The main objective was to examine the validity and relevance of the QST methods in patients with PPSP following GHR. In order to illustrate the findings and characteristics of each study in a systematic manner, both tables and textual references are utilized.”

C14R1: Page 12, line 264: Why did the authors choose to only present log data for MDT and MPT? I would say that data presentation should be homogenous.

A14R1: The authors decided to present the data logarithmically transformed for MDT and MPT, enhancing the clarity and comparability of the results. Raw data for MDT and MPT span a wide range (typically 0.1 – 3,000 mN), often causing skewed data distributions. The ordinal values designated in the monofilament system are logarithmically based (the highest nominal value of the monofilament set used is 6.65, corresponding to a ‘bending weight' of 106.65 (unit 0.1 mg) ~ 447 g or a ‘bending force’ of 4,386 mN) [15].

C15R1: Page 12, line 274: Please reconsider the word “arbitrarily”. If there was a specific intention behind the choice of this value, it does not seem to have been arbitrary.

A15R1: We agree – the word “arbitrarily” has been deleted from the sentence 

C16R1: Page 13, line 290-293: In this section, it is described that twenty studies where non-RCTs and five of those were cross-sectional. What about the other 15?

A16R1: The remaining 15 non-RCTs are classified as cohort studies, as shown in Table 1. We have added the following (page 14, lines 306-307): “The other non-RCTs were cohort studies…….”. 

C17R1: Page 28, line 386: Allodynia would be better described as mechanical hypersensitivity.

A17R1: Thank you for the suggestion. We have added “mechanical hypersensitivity” (page 28, line 400. 

C18R1: Page 30, line 229: What are these values?: (1.6, 6.9)

A18R1: We are grateful for the comment – these are the back-transformed values, now mentioned as following (page 30, lines 463-464): “The total mean MDT difference (back-transformed) in patients with PPSP was 3.3 (1.6, 6.9) mN”. 

Please, observe data in the brackets indicate 95% CI (cf. para. 2.5, line 289).

C19R1: Page 42, line 529-532: In the methods section, the authors mention that for RCTs, only pretreatment QST will be considered. However, in this paragraph, they report decreases in pain in the groups receiving treatment. Wouldn’t it be more relevant to present pre-treatment comparisons? The aim here is not to present QST changes after treatment. The same comment applies for results concerning PPT.

A19R1: For the statistical analysis, we have consistently used pre-treatment data. The mentioned paragraph was solely a narrative synthesis, as the mentioned RCT [9] did not include pre-treatment data in their manuscript. Two RCTs [16, 17] were used in the statistical analysis, and the pre-treatment data were obtained through one of the authors (MW), who was a senior author involved with these studies, as mentioned on page 23, lines 353-356.

C20R1: Page 43, line 570: There should be a summary table of NOS bias assessment in the main manuscript.

A20R1: The comment is appreciated. A summary of the NOS bias assessments is now included (Table 4; page 43, line 587). 

C21R1: Page 44, line 580: The first sentence is too long. Please consider rewrite. Al-so, I would suggest replacing the term “study subjects” by “patients”.

A21R1: We have now rephrased the paragraph as follows (page 45, lines 599-602): “The aims of this systematic review were, first, to identify and describe the available literature on the use of QST in patients with PPSP following GHR, and, second, to explore the role of QST in understanding mechanisms underlying PPSP following GHR.” 

C22R1: Page 44, lines 596-611: The considerations made here concerning RoB are not adequate for a scientific discussion.

A22R1: The Discussion paragraph regarding RoB has now been deleted.

C23R1: Page 46, line 643: Please consider if it would be more correct to state “near-significant difference” instead of “change”. There is no change being assessed, only a side-to-side comparison.

A23R1: Thank you for the observation – we have rephrased the term to “near-significant difference” (page 47, lines 652-653).

C24R1: Page 46, line 651-655: Can the authors explain why this reduction in pain intensity is worth mentioning?

A24R1: We believe it is important since the study [18] represents outlier data (Fig. 2, B). In most studies in persistent postsurgical pain, cutaneous somatosensory assessments are characterized by “deafferentation” or no change in sensory thresholds. Data indicating increased cutaneous sensitivity, i.e., lowered thresh-olds, as in this study [18], are therefore interesting to analyze.

C25R1: Page 47, line 680: Can the authors further explore the meaning of these re-sults? Otherwise, this section is a simple statement of results. The same applies to the discussion of suprathreshold heat stimulus.

A25R1: We have provided a more thorough discussion of the result in section “4.5: Cutaneous vs deep mechanical thresholds”. In regard to STHS, it is men-tioned that the difference in STHS, between the surgical and contralateral side, could be explained by central sensitization phenomenon (page 49, lines 715-717) 

C26R1: Page 48, line 681: There is no discussion of results for WDT, CDT, HPT and CPT.

A26R1: We have now provided a discussion of the thermal thresholds in sections 4.4.2., 4.4.3. and 4.5. 

“4.4.2 . Detection thresholds

Our analyses of thermal detection thresholds showed significant increases in WDT and absolute values of CDT, indicating a loss of small fiber sensory function (cf. section 4.5.).

4.4.3. Pain thresholds

Our analyses of thermal pain thresholds showed a significant increase in HPT, indicating a loss of small fiber sensory function, while no significant differences were found regarding CPT (cf. section 4.5.).

4.5. Cutaneous vs. deep mechanical thresholds

Studies have shown a loss of intraepidermal nerve fiber density (IENFD) on the surgical side in groin hernia repair patients when compared with the contralateral, healthy groin [29, 30]. This decrease in IENFD could serve as an explanation for the hyposensitivity in WDT, CDT, HPT, and MDT. However, no significant differ-ences in sensitivities were found for CPT and MPT*, in spite of a loss of small fi-ber sensory function (theoretically resulting in increased thresholds). This para-doxical finding indicates the presence of a compensatory central sensitization phe-nomenon.”

*Near-significance influenced by a single study (cf. section 4.3.1.)

C1R2: Page 28/58 – “MDT and MPT were included as QST-modalities in one RCT using 17 progressively rigid polyamide monofilaments ranging in bending force from 0.1 to 2941 mN” – were all of them von Frey filaments? Did their use for getting me-chanical thresholds follow the up-and-down method by Chaplan et al 1994? Please explain better.

A1R2: All descriptive terms regarding the monofilaments and the methods de-scribed in the retrieved studies were presented in the review. The up-and-down methods were not consistently presented, but Dixon’s method [19] was mentioned in a single study [12]

C2R2: A high level of heterogeneity was present in several evaluations. How might this interfere with a good evaluation of the results?

A2R2: We have addressed the issue of heterogeneity and its ramifications in sec-tion “4.6 Limitations”, subsection Level of heterogeneity (pages 52-53, lines 789-804)

References:

1. Werner MU, Jensen EK. The Harald Breivik lecture 2022. Pathophysiology in persistent severe pain after groin hernia repair. Scand J Pain. 2022;22(4):686-9.

2. Aiolfi A, Cavalli M, Ferraro SD, Manfredini L, Bonitta G, Bruni PG, et al. Treatment of Inguinal Hernia: Systematic Review and Updated Network Meta-analysis of Randomized Controlled Trials. Ann Surg. 2021;274(6):954-61.

3. Fadaee N, Mazer L, Sharma R, Capati I, Balzer B, Towfigh S. Clinical Value of Hernia Mesh Pathology Evaluation. J Am Coll Surg. 2019;228(5):776-81.

4. Bendavid R, Lou W, Grischkan D, Koch A, Petersen K, Morrison J, et al. A mechanism of mesh-related post-herniorrhaphy neuralgia. Hernia. 2016;20(3):357-65.

5. Iakovlev V, Koch A, Petersen K, Morrison J, Grischkan D, Oprea V, et al. A Pathology of Mesh and Time: Dysejaculation, Sexual Pain, and Orchialgia Resulting From Polypropylene Mesh Erosion Into the Spermatic Cord. Ann Surg. 2018;267(3):569-75.

6. Jensen EK, Ringsted TK, Bischoff JM, Petersen MA, Rosenberg J, Kehlet H, et al. A national center for persistent severe pain after groin hernia repair: Five-year prospective data. Medicine (Baltimore). 2019;98(33):e16600.

7. Amid PK. Radiologic images of meshoma: a new phenomenon causing chronic pain after prosthetic repair of abdominal wall hernias. Arch Surg. 2004;139(12):1297-8.

8. Bischoff JM, Koscielniak-Nielsen ZJ, Kehlet H, Werner MU. Ultrasound-guided ilioinguinal/iliohypogastric nerve blocks for persistent inguinal postherniorrhaphy pain: a randomized, double-blind, placebo-controlled, crossover trial. Anesth Analg. 2012;114(6):1323-9.

9. Wijayasinghe N, Ringsted TK, Bischoff JM, Kehlet H, Werner MU. The role of peripheral afferents in persistent inguinal postherniorrhaphy pain: a randomized, double-blind, placebo-controlled, crossover trial of ultrasound-guided tender point blockade. Br J Anaesth. 2016;116(6):829-37.

10. Aasvang EK, Kehlet H. The effect of mesh removal and selective neurectomy on persistent postherniotomy pain. Ann Surg. 2009;249(2):327-34.

11. Beel E, Berrevoet F. Surgical treatment for chronic pain after inguinal hernia repair: a systematic literature review. Langenbecks Arch Surg. 2022;407(2):541-8.

12. Jensen EK, Ringsted TK, Bischoff JM, Petersen MA, Møller K, Kehlet H, et al. Somatosensory Outcomes Following Re-Surgery in Persistent Severe Pain After Groin Hernia Repair: A Prospective Observational Study. J Pain Res. 2023;16:943-59.

13. Schug SA, Lavand'homme P, Barke A, Korwisi B, Rief W, Treede RD. The IASP classification of chronic pain for ICD-11: chronic postsurgical or posttraumatic pain. Pain. 2019;160(1):45-52.

14. Kristensen AD, Ahlburg P, Lauridsen MC, Jensen TS, Nikolajsen L. Chronic pain after inguinal hernia repair in children. Br J Anaesth. 2012;109(4):603-8.

15. Ringsted TK, Enghuus C, Petersen MA, Werner MU. Demarcation of secondary hyperalgesia zones: Punctate stimulation pressure matters. J Neurosci Methods. 2015;256:74-81.

16. Bischoff JM, Ringsted TK, Petersen M, Sommer C, Uçeyler N, Werner MU. A capsaicin (8%) patch in the treatment of severe persistent inguinal postherniorrhaphy pain: a randomized, double-blind, placebo-controlled trial. PLoS One. 2014;9(10):e109144.

17. Bischoff JM, Petersen M, Uçeyler N, Sommer C, Kehlet H, Werner MU. Lidocaine patch (5%) in treatment of persistent inguinal postherniorrhaphy pain: a randomized, double-blind, placebo-controlled, crossover trial. Anesthesiology. 2013;119(6):1444-52.

18. Bjurström MF, Nicol AL, Amid PK, Lee CH, Ferrante FM, Chen DC. Neurophysiological and Clinical Effects of Laparoscopic Retroperitoneal Triple Neurectomy in Patients with Refractory Postherniorrhaphy Neuropathic Inguinodynia. Pain Pract. 2017;17(4):447-59.

19. Dixon WJ. Staircase bioassay: the up-and-down method. Neurosci Biobehav Rev. 1991;15(1):47-50.

---

## [Decision Letter · Decision Letter 1]

5 Sep 2023

PONE-D-23-14833R1Quantitative somatosensory assessments in patients with persistent pain following groin hernia repar: A systematic review with a meta-analytical approachPLOS ONE

Dear Dr. Dubayev,

Thank you for submitting your manuscript to PLOS ONE. After careful consideration, we feel that it has merit but does not fully meet PLOS ONE’s publication criteria as it currently stands. Therefore, we invite you to submit a revised version of the manuscript that addresses the points raised during the review process.

We look forward to receiving your revised manuscript.

Kind regards,

Armando Almeida

Academic Editor

PLOS ONE

Journal Requirements:

**Additional Editor Comments:**

While Reviewer #2 already accepted the manuscript as it is, Reviewer #1 has still a few comments to be addressed by the authors.

Reviewers' comments:

Reviewer's Responses to Questions

**Comments to the Author**

1. If the authors have adequately addressed your comments raised in a previous round of review and you feel that this manuscript is now acceptable for publication, you may indicate that here to bypass the “Comments to the Author” section, enter your conflict of interest statement in the “Confidential to Editor” section, and submit your "Accept" recommendation.

Reviewer #1: (No Response)

Reviewer #2: All comments have been addressed

2. Is the manuscript technically sound, and do the data support the conclusions?

Reviewer #1: Yes

Reviewer #2: (No Response)

3. Has the statistical analysis been performed appropriately and rigorously? 

Reviewer #1: Yes

Reviewer #2: (No Response)

4. Have the authors made all data underlying the findings in their manuscript fully available?

Reviewer #1: Yes

Reviewer #2: (No Response)

5. Is the manuscript presented in an intelligible fashion and written in standard English?

Reviewer #1: Yes

Reviewer #2: (No Response)

6. Review Comments to the Author

Reviewer #1: I thank the authors for all their careful revisions.

I have some additional comments below. Page and line numbers are from the version with tracked changes.

Page 8, lines 89-92: Despite the changes made by the authors, I still struggle to understand how this work will result in “enhanced prediction of prognostic outcomes”. By prognostic outcomes do you mean the QST? I would suggest to delete this sentence.

Page 10, lines 211-213: Beware that this sentence is repeated below (in red).

Page 10, line 218: Thank you for this clarification. I would just advise to clarify if it is “and” or “or”, and eliminate the “and/or” option. I say this because if you write “at least one standard mechanical AND one thermal”, it means that both had to be present in the paper. On the other hand, if you write “at least one standard mechanical OR one thermal”, it means that the study would be included if only one QST modality was present. Since the sentences have very different meanings, “and/or” is not adequate. Please, select one.

Eligibility criteria: Please, also mention in this section that there were no restrictions in terms of age. Also, it should be clearly mentioned here that the included studies examined the respective cohorts after a post-surgical time period of at least three months.

Page 14, line 309: I think that the analyses could be better understood if the authors mentioned here the calculations performed (back-transformed mean differences). This would clarify which values were included in the forest plots.

Page 30, line 319: The authors could write here “decrease in threshold from the affected side to control side”.

Page 30, lines 463-464: If the information in brackets are the CI, this should be in the brackets (95% CI=1.6, 6.9).

Page 46, line 637: The second aim is very clearly stated here. I suggest writing it the same way in the abstract and aim sections.

Page 47, line 664: I suggest presenting all the discussion for RoB under the subtitle 4.2. Otherwise, it is strange why the Cochrane Tool is not mentioned.

Page 54, line 836: I suggest adding some comment about the possible limitation of making comparisons with the contralateral site in some studies, but with healthy subjects in other studies.

Reviewer #2: (No Response)

7. PLOS authors have the option to publish the peer review history of their article (what does this mean?). If published, this will include your full peer review and any attached files.

Reviewer #1: No

Reviewer #2: No

---

## [Author Response · Author response to Decision Letter 1]

26 Sep 2023

The following is also uploaded as a Word document titled "Response_to_reviewers_R2":

PONE-D-23-14833R1

Response to PLOS ONE second revision letter 

Quantitative somatosensory assessments in patients with persistent pain fol-lowing groin hernia repair: A systematic review with a meta-analytical ap-proach

Dear Academic Editor Armando Almeida, 

We would like to express our gratitude for the opportunity to further revise and resubmit our manuscript to PLOS ONE. We are grateful that our work is being considered for publication in your journal.

We appreciate the careful review and constructive feedback provided by the re-viewers. Your insights have been invaluable in improving the quality and clarity of our study. 

Please, find below our replies to the Reviewers' comments. The Reviewers' com-ments are numbered chronologically (C1, C2, C3…) and according to the Review-er’s # (R1, R2) followed by the authors' answers (A1, A2...). Page and line num-bers refer to the version with track changes.

Best regards on behalf of the authors,

Akhmedkhan Dubayev, M.D., and Mads U. Werner, M.D., Ph.D.

REVIEWER #1

C1R1: Page 8, lines 88-92: Despite the changes made by the authors, I still struggle to understand how this work will result in “enhanced prediction of prognostic out-comes”. By prognostic outcomes do you mean the QST? I would suggest to delete this sentence.

A1R1: Thank you for the suggestion. We have deleted the sentence. 

C2R1: Page 10, lines 211-213: Beware that this sentence is repeated below (in red).

A2R1: Thank you for noticing the error. We have now deleted one of the sentenc-es. 

C3R1: Page 10, line 218: Thank you for this clarification. I would just advise to clari-fy if it is “and” or “or”, and eliminate the “and/or” option. I say this because if you write “at least one standard mechanical AND one thermal”, it means that both had to be present in the paper. On the other hand, if you write “at least one standard me-chanical OR one thermal”, it means that the study would be included if only one QST modality was present. Since the sentences have very different meanings, “and/or” is not adequate. Please, select one.

Eligibility criteria: Please, also mention in this section that there were no restrictions in terms of age. Also, it should be clearly mentioned here that the included studies examined the respective cohorts after a post-surgical time period of at least three months.

A3R1: We thank you for the comment. We agree, and we have changed it to “or” clarifying our standpoint. 

C4R1: Page 14, line 309: I think that the analyses could be better understood if the authors mentioned here the calculations performed (back-transformed mean differ-ences). This would clarify which values were included in the forest plots.

A4R1: Transformed values are indicated in the tabulated part of the forest plots. MDT and MPT values are not back-transformed in the forest plots. Back transfor-mations are solely provided in the main text (page 30, line 469 and page 40, line 496).

C5R1: Page 30, line 319: The authors could write here “decrease in threshold from the affected side to control side”.

A5R1: The sentence has been rephrased to accommodate the suggestion. 

C6R1: Page 30, lines 463-464: If the information in brackets are the CI, this should be in the brackets (95% CI=1.6, 6.9)

A6R1: “95% CI” has now been added in the brackets in our Results’ section. 

C7R1: Page 46, line 637: The second aim is very clearly stated here. I suggest writing it the same way in the abstract and aim sections.

A7R1: We thank you for the comment. We have rephrased the second aim in the Abstract and the Aims (1.1.3.) section (page 4, lines 76-77 and page 8, line 168-169). 

C8R1: Page 47, line 664: I suggest presenting all the discussion for RoB under the subtitle 4.2. Otherwise, it is strange why the Cochrane Tool is not mentioned.

A8R1: The discussion for RoB 2.0 has now been reinstated under section 4.2. 

C9R1: Page 54, line 836: I suggest adding some comment about the possible limita-tion of making comparisons with the contralateral site in some studies, but with healthy subjects in other studies.

A9R1: We are grateful for the suggestion. We have written the following para-graph to comment on the issue in subsection 4.6.1. Studies (page 55, lines 842-854): 

“Healthy controls vs. contralateral side: The controlled studies used different meth-odological approaches when comparing the sensory abnormalities at the surgical site to a control site. Either an absolute approach, comparing with a normative healthy cohort, or a relative approach, comparing with the individual’s contralateral homotopic site, or a combination of these approaches, were used. One of the ad-vantages of the relative approach is that the within-subject variances often are significantly smaller than the between-subject variances. Using the individual’s contralateral site as a control is thus expected to reduce data variability, making the data more robust and less susceptible to confounding factors such as age, gender, and random errors. On the other hand, mirror-image sensory dysfunction [1], a neural cross-talk between the sides, may influence the side-to-side differ-ence in the relative approach. Very few studies have systematically examined the pros and cons of the absolute and relative approaches [2].”

References

1. Enax-Krumova E, Attal N, Bouhassira D, Freynhagen R, Gierthmühlen J, Hansson P, et al. Contralateral Sensory and Pain Perception Changes in Patients With Unilateral Neuropathy. Neurology. 2021;97(4):e389-e402.

2. Kemler M, Schouten H, Gracely R. Diagnosing Sensory Abnormalities with Either Normal Values or Values from Contralateral Skin. Anesthesiology. 2000;93:718-27.

---

## [Editor Report · Decision Letter 2]

29 Sep 2023

Quantitative somatosensory assessments in patients with persistent pain following groin hernia repar: A systematic review with a meta-analytical approach

PONE-D-23-14833R2

Dear Dr. Dubayev,

We’re pleased to inform you that your manuscript has been judged scientifically suitable for publication and will be formally accepted for publication once it meets all outstanding technical requirements.

Kind regards,

Armando Almeida

Academic Editor

PLOS ONE
---

## [Editor Report · Acceptance letter]

16 Oct 2023

PONE-D-23-14833R2 

Quantitative somatosensory assessments in patients with persistent pain following groin hernia repair: A systematic review with a meta-analytical approach 

Dear Dr. Dubayev:

I'm pleased to inform you that your manuscript has been deemed suitable for publication in PLOS ONE. Congratulations! Your manuscript is now with our production department. 

Kind regards, 

on behalf of

Prof. Armando Almeida 

Academic Editor

PLOS ONE